# Increased expression of heme-binding protein 1 early in Alzheimer's disease is linked to neurotoxicity

Oleksandr Yagensky[1]*, Mahdokht Kohansal-Nodehi[2†], Saravanan Gunaseelan[3†], Tamara Rabe[4], Saima Zafar[5,6], Inga Zerr[6], Wolfgang Härtig[7], Henning Urlaub[8,9], John JE Chua[1,3,10,11,12]*

[1]Research Group Protein Trafficking in Synaptic Development and Function, Max Planck Institute for Biophysical Chemistry, Göttingen, Germany; [2]Department of Neurobiology, Max Planck Institute for Biophysical Chemistry, Göttingen, Germany; [3]Interactomics and Intracellular Trafficking Laboratory, Department of Physiology, Yong Loo Lin School of Medicine, National University of Singapore, Singapore, Singapore; [4]Department of Genes and Behavior, Max Planck Institute for Biophysical Chemistry, Göttingen, Germany; [5]Biomedical Engineering and Sciences Department, School of Mechanical and Manufacturing Engineering (SMME), National University of Sciences and Technology (NUST), Islamabad, Pakistan; [6]Clinical Dementia Center, Department of Neurology, German Center for Neurodegenerative Diseases, University Medical Center Göttingen, Göttingen, Germany; [7]Paul Flechsig Institute for Brain Research, University of Leipzig, Leipzig, Germany; [8]Research Group Bioanalytical Mass Spectrometry, Max Planck Institute for Biophysical Chemistry, Göttingen, Germany; [9]Bioanalytics Group, Institute for Clinical Chemistry, University Medical Center Göttingen, Göttingen, Germany; [10]LSI Neurobiology Programme, National University of Singapore, Singapore, Singapore; [11]Institute of Molecular and Cell Biology, Agency for Science, Technology and Research (A*STAR), Singapore, Singapore; [12]Institute for Health Innovation and Technology, National University of Singapore, Singapore, Singapore

*For correspondence:
oleksandr.yagensky@mpibpc.mpg.de (OY);
phsjcje@nus.edu.sg (JJEC)

†These authors contributed equally to this work

Competing interests: The authors declare that no competing interests exist.

**Abstract** Alzheimer's disease is the most prevalent neurodegenerative disorder leading to progressive cognitive decline. Despite decades of research, understanding AD progression at the molecular level, especially at its early stages, remains elusive. Here, we identified several presymptomatic AD markers by investigating brain proteome changes over the course of neurodegeneration in a transgenic mouse model of AD (3×Tg-AD). We show that one of these markers, heme-binding protein 1 (Hebp1), is elevated in the brains of both 3×Tg-AD mice and patients affected by rapidly-progressing forms of AD. Hebp1, predominantly expressed in neurons, interacts with the mitochondrial contact site complex (MICOS) and exhibits a perimitochondrial localization. Strikingly, wildtype, but not Hebp1-deficient, neurons showed elevated cytotoxicity in response to heme-induced apoptosis. Increased survivability in Hebp1-deficient neurons is conferred by blocking the activation of the mitochondrial-associated caspase signaling pathway. Taken together, our data highlight a role of Hebp1 in progressive neuronal loss during AD progression.
DOI: https://doi.org/10.7554/eLife.47498.001

## Introduction

Alzheimer's disease is a progressive neurodegenerative disorder that leads to memory loss and cognitive decline. It is the most prevalent form of dementia in the elderly and is projected to affect more than 40 million people worldwide (*GBD , Disease and Injury Incidence and Prevalence Collaborators et al., 2017*). At the molecular level, AD is characterized by a disturbed metabolism of amyloid beta (Aβ) peptides that results in formation of toxic oligomers and insoluble aggregates (plaques) in the brains of afflicted individuals (*Selkoe and Hardy, 2016*). AD pathology is also accompanied by formation of neurofibrillary tangles comprising of hyperphosphorylated microtubule-associated protein tau. Aβ deposits and phospho-tau-containing neurofibrils serve as molecular hallmarks of AD and are thus useful for histopathological diagnosis (*Vinters, 2015*). However, the aggregation of Aβ and tau alone does not fully account for the cognitive decline observed in AD patients (*Davis et al., 1999*). Although mutations in amyloid precursor protein (APP) and its proteases presenilin-1 and 2 have a causative relationship with the onset of familiar form of Alzheimer's disease (FAD), overt manifestation of clinical AD is often preceded by a prolonged incubation period (*Ryman et al., 2014*). This leads to recognition of AD as a complex multifaceted disorder that strongly depends on the intricate interplay between neuronal survival, synaptic function, activation of glial cells, inflammatory response, blood-brain barrier impairment and other factors (*De Strooper and Karran, 2016*). Nevertheless, the knowledge of the biological processes that are first affected in AD remain limited. Identification of these processes will expand our understanding of early AD pathogenesis and may lay the ground for the development of more effective therapeutics in the future.

Changes in gene expression can be indicative of underlying physiological and pathological alterations during disease progression. Indeed, studies utilizing cDNA microarray, RNA sequencing and mass spectrometry approaches to analyze such changes in human postmortem brain tissue have broadened our understanding of genes and proteins involved in AD (*Alkallas et al., 2017*; *Donovan et al., 2012*; *Mathys et al., 2017*; *Moya-Alvarado et al., 2016*; *Musunuri et al., 2014*; *Podtelezhnikov et al., 2011*). While such studies provide important insights into molecular pathology at later stages, they offer limited information about prior alterations that occur over the course of the development of the disorder. In particular, early changes in protein expression preceding the clinical onset of the disease would be missed. Detection of such presymptomatic protein markers would not only aid the earlier diagnosis of afflicted individuals but also potentially enable the identification of early targets for therapeutic intervention.

To identify such markers, we probed for temporal changes in the brain proteome of 3×Tg-AD transgenic mice that harbor three mutated genes associated with the disease (*PSEN1* M146V, *APP* Swe, *MAPT* (tau) P301L). These mice develop Alzheimer-related phenotypes in a progressive manner mimicking the human disorder (*Oddo et al., 2003a*; *Oddo et al., 2003b*). Quantitative mass spectrometry was employed to compare the brain proteomes of 3×Tg-AD transgenic mice against age-matched controls at four time points corresponding to various stages of the disorder. Both age- and disease-dependent alterations could be observed in the brain proteome of 3×Tg-AD mice. Significantly, the analyses further revealed several potential presymptomatic protein markers that are differentially expressed between 3×Tg-AD and control mice. One of these markers, heme-binding protein-1 (Hebp1), is significantly elevated in the brains of both 3×Tg-AD mice and human patients exhibiting rapidly-progressing forms of AD. Hebp1 is primarily expressed in neurons where it is associated with mitochondria via the MICOS complex. Functionally, Hebp1 mediates heme-induced cytotoxicity via an apoptotic pathway. Thus, it is of relevance both as an early marker and contributing factor to the development of AD.

## Results

### Brain proteomes of wild-type and 3×Tg-AD mice exhibit age- and disease-related changes

To identify proteins involved in early stage AD, we monitored for changes in the brain proteomes of control and 3×Tg-AD mice at four different time points using label-free quantitative mass spectrometry. These time points were selected according to the pathological changes in 3×Tg-AD mice based on previously published data (*Hawkes et al., 2013*; *Oddo et al., 2003a*; *Wirths et al., 2012*) and our own observations (*Figure 1—figure supplement 1*). We included presymptomatic time

point (2 months), the age of first behavioral abnormalities (6 months), appearance of first Aβ plaques and hyperphosphorylated tau (12–18 months) (*Figure 1A*). At the designated time points, one half of the harvested brain from each animal was processed to obtain a soluble protein fraction that was subjected to analyses by label-free liquid chromatography-tandem mass spectrometry (LC-MS/MS)

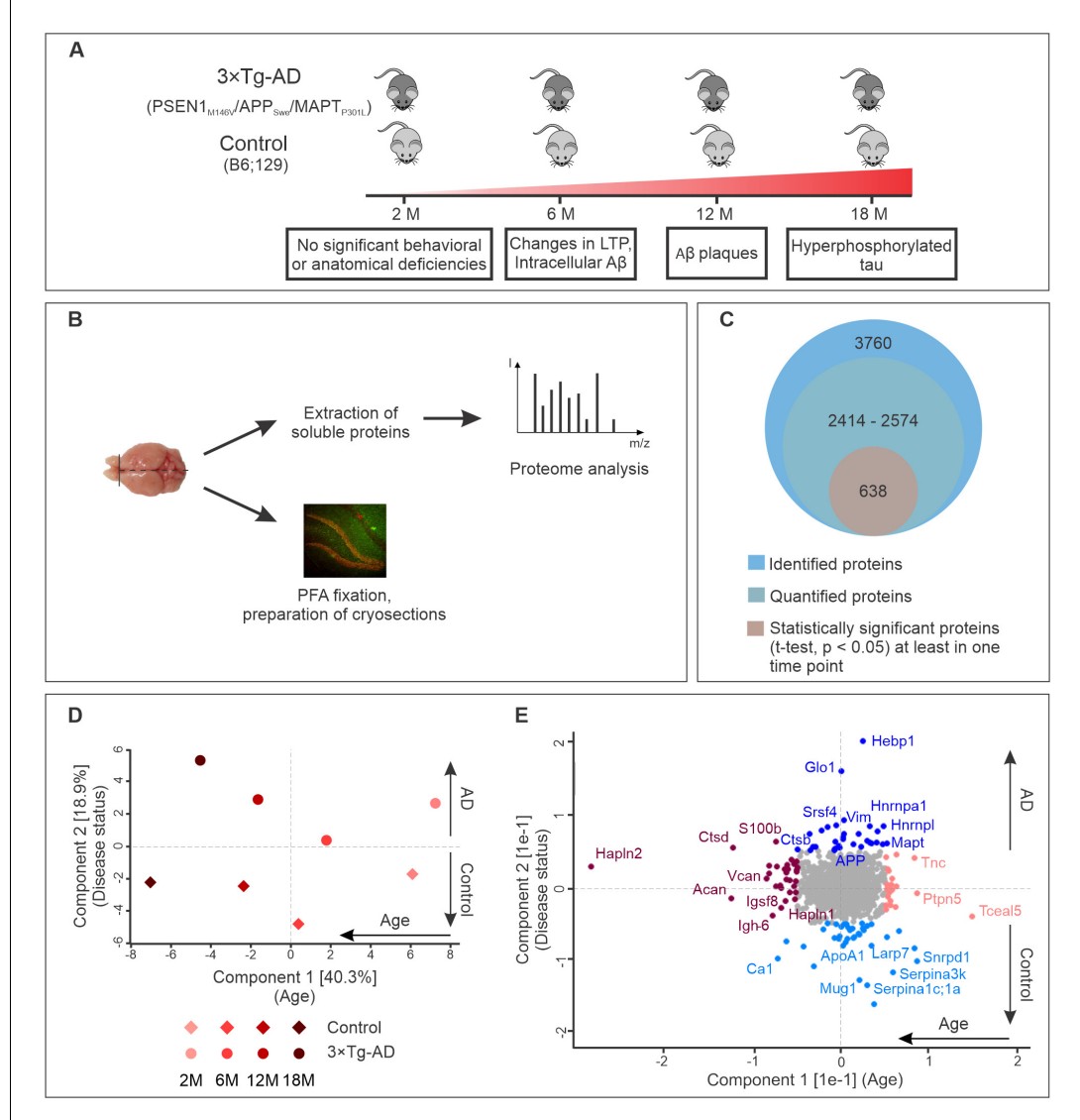

**Figure 1.** Progression of Alzheimer's disease at molecular level in the triple transgenic mouse model (PSEN1$_{M146V}$/APP$_{Swe}$/MAPT$_{P301L}$). (**A**) Disease progression in 3×Tg-AD mice and corresponding time points (2, 6, 12, 18 months) of sample collection. Four biological replicates per group were collected at each time point. (**B**) Experimental workflow and sample processing. Half of the collected brain sample was used for preparation of the cryosections for immunohistochemistry. Soluble proteins of the other half were extracted for proteomics analysis. (**C**) Number of identified, quantified and statistically significant proteins in the dataset. (**D**) Principal component analysis of soluble brain proteome of 3×Tg-AD and control mice based on their protein expression profile. Principal component one segregates mice by age and accounts for 40.3% of variability in the dataset, while principal component two clusters mice according to their disease status (18.9% of variance). (**E**) Proteins driving the differences in proteomes between aged, young, diseased and control mice depicted in brown, pink, navy and light blue colors, respectively.

DOI: https://doi.org/10.7554/eLife.47498.002

The following figure supplements are available for figure 1:

**Figure supplement 1.** Assessment of pathology in 3×Tg-AD mice used in this study.
DOI: https://doi.org/10.7554/eLife.47498.003

**Figure supplement 2.** Technical reproducibility between biological replicates.
DOI: https://doi.org/10.7554/eLife.47498.004

(*Figure 1B*). Cryosections were prepared from the other half of the brain for subsequent immunohistochemical analyses of the hits identified by mass spectrometry.

Analysis of the datasets using MaxQuant software (*Cox and Mann, 2008*), applying a peptide and protein false discovery rate (FDR) of 1%, identified a total of 3760 protein groups in the soluble brain protein fraction (*Figure 1C*). Of these, only proteins that were identified in at least two out of four biological replicates for each group (disease or control) at each time point were used for further analyses. In this way, between 2414 and 2574 proteins were quantified depending on the time point. Pearson's correlation coefficients for quantified proteins between biological replicates were above 0.96 attesting to the high reproducibility of the data (*Figure 1—figure supplement 2*).

Principal component analysis of the datasets revealed that these proteins can be clustered according to age (component 1, 40.9% of total variation) as well as disease state (component 2, 18.3% of total variation) (*Figure 1D*). Notably, increasing difference between control and AD brain proteome could be observed with aging and disease progression (*Figure 1D*). The segregation by age was mainly driven by extracellular matrix proteins (Hapln2, Tnc, Acan, Vcan, Hapln1) and increased expression of microglia markers (S100b, Ctpd) (*Figure 1E*). Many of these proteins have been previously reported as markers of brain aging (*Sato and Endo, 2010*; *Végh et al., 2014*). The samples segregated by principle component two varied primarily in the expression of AD-related genes. As expected, APP and tau (MAPT) were clustered together with proteins upregulated in AD (*Figure 1E*).

## Relating proteome changes to biological processes in AD progression

To determine which biological processes were significantly affected with relation to disease progression, we subjected the dataset to Ingenuity Pathway Analysis (IPA) (QIAGEN Inc, https://www.qiagenbioinformatics.com/products/ingenuity-pathway-analysis). We grouped the processes based on the trend of their activation z-score at early (6 months, *Figure 2A*), intermediate (12 months, *Figure 2B*) and late symptomatic time points (18 months, *Figure 2C*). Remarkably, cumulative upregulation in expression of proteins involved in cell death and apoptotic processes could already be detected at the first symptomatic time point (6 months) (*Figure 2A*). Signatures of mitochondria dysfunction were also among the very first signs of altered AD proteome (*Figure 2A,B*). Significant changes in regulation of proteins associated with seizures were also observed at the transition point between presymptomatic phase and 6 months which corresponds to the stage where changes in long-term potentiation (LTP) in 3×Tg-AD mice were previously described (*Oddo et al., 2003b*; *Palop et al., 2007*). Neurodegeneration-related processes (amyloid load of hippocampus, demyelination of axons and degradation of mitochondria) were also noticeably exacerbated with AD progression (*Figure 2B,C*). Notably, disturbance of the cytoskeleton, which is a hallmark of many neurodegenerative disorders including AD, became prominent only at the late stage of the disorder (18 months) (*Figure 2C*) and correlated with the emergence of hyperphosphorylated tau in 3×Tg-AD mice (*Figure 1—figure supplement 1*).

To identify the primary drivers of the phenotypes observed at each stage, identification of top upstream regulators was performed using IPA. This revealed that mutant tau, APP and PSEN1 were principally responsible for the differences observed between wild-type and 3×Tg-AD brains (*Figure 2—figure supplement 1A*). Interestingly, the proteins that are regulated by tau, APP and PSEN1 in our dataset largely overlap (*Figure 2—figure supplement 1B*). Protein enrichment by biological function indicates that these downstream effectors contribute primarily to apoptosis, mitochondria dysfunction and oxidative stress (*Figure 2—figure supplement 1C*).

To identify potential markers indicative of disease onset and progression, we scanned the dataset for proteins with the highest degree of fold-change at each AD stage. Therefore, proteins demonstrating more than 50% change in expression level between disease and control that was statistically significant (t-test, p<0.05) were shortlisted (*Figure 2D–G*). Among these, groups of proteins involved in the same biological function could be identified. For instance, several proteins involved in mRNA processing (Hnrnpm, Hnrnpl, Nono, Matrin3) were strongly upregulated at the later time points. Remarkably, expression of these proteins increased gradually in coordinated manner throughout the progression of the disorder (*Figure 2H*). A similar coordinated expression pattern was also observed for the group of serine protease inhibitors (Serpina1c, Serpina3k, Mug1) which were significantly downregulated by the latest time point examined (*Figure 2I*). This result is particularly interesting in light of recent findings linking a reduction in protease inhibitors levels to aging and cognitive decline

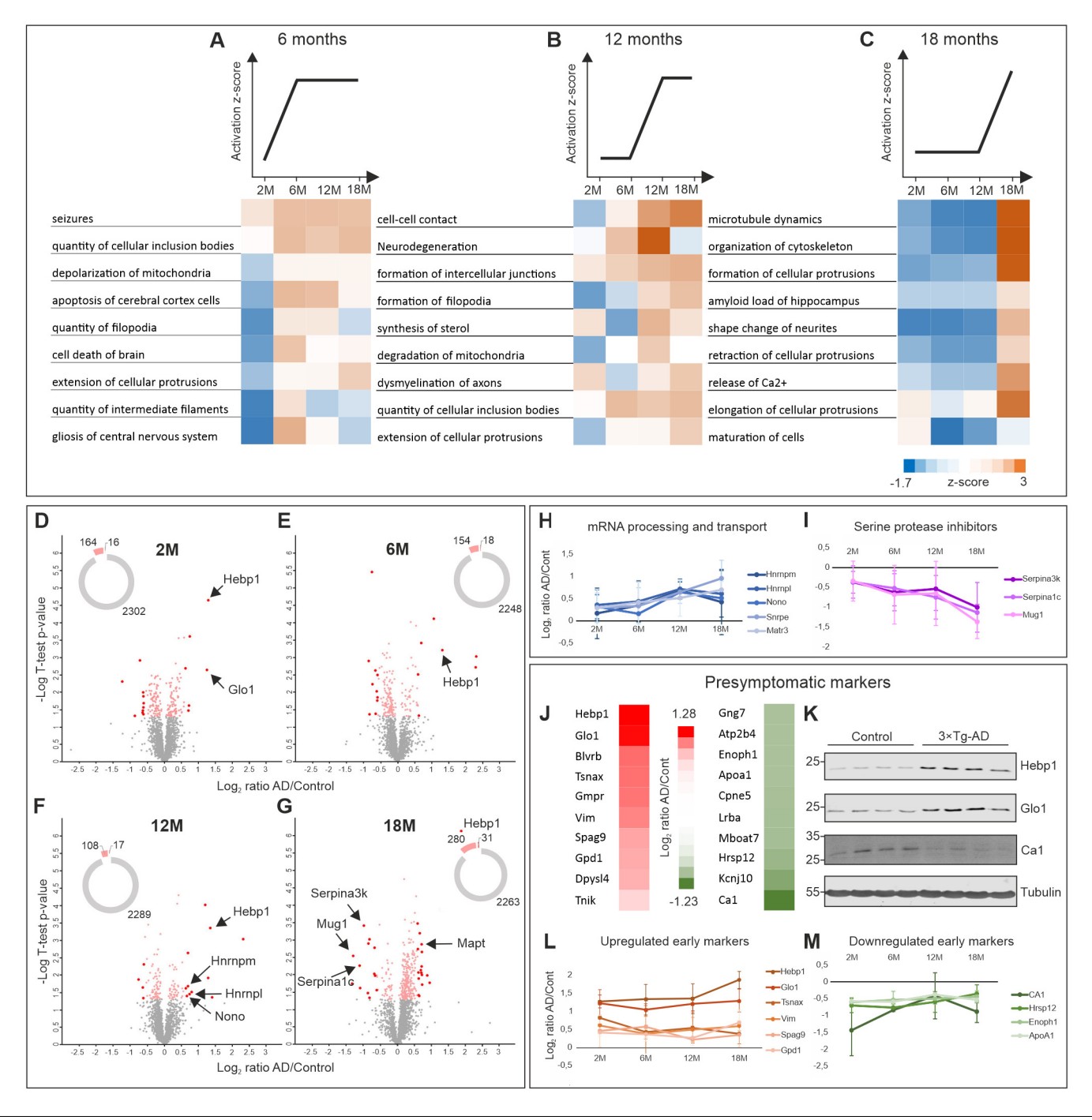

**Figure 2.** Comparative proteome analysis of 3×Tg-AD and control samples at different stages of AD. (A, B, C) Activation of biological processes at different stages of the disease assessed by Ingenuity Pathway Analysis (IPA). Heat maps represent activation z-score change over the course of disease progression and indicate pathways that are activated at 6M (A), 12M (B) and 18M (C). Data were obtained from four biological replicates per group for each time point. Z-score is calculated based on experimental protein expression data ($\log_2$ AD/control ratio) and the theoretical information stored in the IPA Knowledge Base. Positive value of z-score indicates an activation of biological pathway or function. Distribution of the quantified proteins at 2M (D), 6M (E), 12M (F) and 18M (G) based on $\log_2$ ratio AD/Control and p-value (t-test) by time point. The pie charts represent the number of quantified non-regulated proteins (grey), significantly different proteins between 3×Tg-AD and control samples, t-test p-value 0.05 (pink) and significantly regulated proteins with more than 50% expression change in comparison to the control (red). (H–I) Dynamics of protein expression over the course of AD progression for a selection of the most regulated proteins based on their function. Proteins involved in mRNA processing and transport (H) that are upregulated over time and serine protease inhibitors (I) that are downregulated. (J–M) Putative presymptomatic protein markers of the disease. (J) Top

*Figure 2 continued on next page*

*Figure 2 continued*

10 significantly up- and downregulated proteins in 3×Tg-AD mice at presymptomatic time point (2M). (K) Immunoblot analysis of most regulated hits. Soluble fractions of brain proteins were analyzed from four 2-month-old control and 3×Tg-AD mice animals, respectively. Hebp1 and Glo1 levels were consistently elevated in the transgenic animals as compared to wild type controls. Ca1 levels were reduced in transgenic animals. Presymptomatic markers that remain up- (L) or downregulated (M) across the AD progression.

DOI: https://doi.org/10.7554/eLife.47498.005

The following source data and figure supplement are available for figure 2:

**Source data 1.** Full list of quantified proteins in the soluble brain fraction of 3×Tg-AD and wild-type mice.

DOI: https://doi.org/10.7554/eLife.47498.007

**Figure supplement 1.** Analysis of upstream regulators.

DOI: https://doi.org/10.7554/eLife.47498.006

(*Castellano et al., 2017*). We have also identified several inflammation-related proteins (C1qc, Ilf2, Igh-3) and components of myelin sheath (Mag, Mog) to be strongly up- or downregulated at the pre-terminal stage of the disorder (18 months). This hints towards progressive inflammation and demyelination in the analyzed 3×Tg-AD model which has been linked to AD (*Wyss-Coray and Rogers, 2012*; *Zhan et al., 2014*).

## Presymptomatic protein markers of AD

We further narrowed our analyses to focus on proteins with the strongest fold changes at the presymptomatic stage for two main reasons. First, these proteins could be useful as potential early markers indicative of disease onset. Second, they might be responsible for causing the initial pathogenic alterations. Thus, deciphering their function can help us better understand the initial cascade of events driving the onset of AD. We identified strongly up- or downregulated proteins at the 2 month time point (*Figure 2J*) and corroborated the most prominent hits by immunoblotting against protein samples obtained from the respective 2-month-old animals (*Figure 2K*). Many of these presymptomatic markers maintained their expression levels across later time points in 3×Tg-AD mice, suggesting their relevance for the late stages of AD as well (*Figure 2L,M*). Noteworthy, half of the identified putative early markers were previously associated with AD or other neurodegenerative disorders (*Table 1*). For example, decreased levels of ApoA1 that we observe in our dataset have been linked to the increased severity of AD in human patients (*Merched et al., 2000*; *Saczynski et al., 2007*). Guanosine monophosphate reductase 1 (Gmpr) which was identified to be expressed at increased levels in the brains of AD patients followed the same direction of change in our dataset (*Liu et al., 2018*). Glyoxalase-1 (Glo1), another protein whose elevated expression was detected in our study, was previously shown to be upregulated in other mouse models of neurodegeneration (*Chen et al., 2004*). Furthermore, restoration of Glo1 activity has been proposed as a mechanism to combat cognitive dysfunction in AD (*More et al., 2013*).

Among the newly identified putative presymptomatic markers of AD, heme-binding protein 1 (Hebp1) is a particularly interesting candidate. In our dataset, it was the most highly and consistently upregulated protein at all time points. Hebp1 belongs to the SOUL protein family and was originally identified as a tetrapyrol-binding protein capable of binding protoporphyrin IX and heme (*Jacob Blackmon et al., 2002*; *Dias et al., 2006*; *Taketani et al., 1998*). Heme is essential for proper mitochondria function and cell survival (*Atamna, 2004*). Impairments in heme metabolism are also associated with AD (*Atamna and Frey, 2004*). Our data show that cell survival and mitochondria function might be among the first pathways affected in AD (*Figure 2A and B*; *Figure 2—figure supplement 1C*). To the best of our knowledge, no information on Hebp1 function in the brain is available to date. We thus further investigated the function of Hebp1 and its potential role in Alzheimer's disease.

## Hebp1 is upregulated in rapidly-progressing cases of human AD

To verify the relevance of our findings in the mouse model for the disorder in humans, we examined the expression of Hebp1 and Glo1, the two most upregulated early markers, in postmortem brain samples obtained from AD patients and age-matched healthy controls (*Figure 3*; detailed patient information is provide in *Table 2*). We could confirm an overall increase in expression of both proteins in AD patients compared to controls that validated our findings obtained in the 3×Tg-AD

**Table 1.** Identified presymptomatic brain markers of AD in this study.

| Gene name | Protein name | Log$_2$ AD/Ctrl | Previous involvement in AD | Reference |
|---|---|---|---|---|
| *Upregulated presymptomatic markers* | | | | |
| Hebp1 | Heme binding protein 1 | 1.28 | - | - |
| Glo1 | Glyoxalase 1 | 1.24 | ↑ in human brain, mouse model of FTD | (*Chen et al., 2004*), (*More et al., 2013*) |
| Blvrb | Biliverdin Reductase B | 0.74 | ↑ in plasma | (*Mueller et al., 2010*) |
| Tsnax | Translin Associated Factor X | 0.72 | - | - |
| Gmpr | Guanosine Monophosphate Reductase | 0.70 | ↑ human brain, early stage | (*Liu et al., 2018*) |
| Vim | Vimentin | 0.62 | ↑ in human brain (astrocytes) | (*Yamada et al., 1992*) |
| Spag9 | Sperm Associated Antigen 9 | 0.46 | - | - |
| Gpd1 | Glycerol-3-Phosphate Dehydrogenase 1 | 0.43 | Accumulation in NFT | (*Wang et al., 2005*) |
| Dpysl4 | Dihydropyrimidinase Like 4 | 0.40 | - | - |
| Tnik | TRAF2 And NCK Interacting Kinase | 0.22 | Accumulation in insoluble fraction of amygdala in cognitively impaired patients | (*Gal et al., 2018*) |
| *Downregulated presymptomatic markers* | | | | |
| Gng7 | G Protein Subunit Gamma 7 | −0.60 | - | - |
| Atp2b4 | ATPase Plasma Membrane Ca$^{2+}$ Transporting 4 | −0.60 | ↓ in human brain | (*Kong et al., 2015*) |
| Enoph1 | Enolase-Phosphatase 1 | −0.60 | - | - |
| Apoa1 | Apolipoprotein A1 | −0.60 | ↓ in plasma | (*Saczynski et al., 2007*), (*Merched et al., 2000*) |
| Cpne5 | Copine 5 | −0.62 | - | - |
| Lrba | LPS Responsive Beige-Like Anchor Protein | −0.63 | - | - |
| Mboat7 | Membrane Bound O-Acyltransferase Domain Containing 7 | −0.63 | - | - |
| Hrsp12 | Ribonuclease UK114 | −0.71 | ↑ in CVN-AD model | (*Hoos et al., 2013*) |
| Kcnj10 | Potassium Voltage-Gated Channel Subfamily J Member 10 | −0.86 | ↓ in mouse model of ALS | (*Kaiser et al., 2006*) |
| Ca1 | Carbonic anhydrase 1 | −1.23 | - | - |

DOI: https://doi.org/10.7554/eLife.47498.008

model. Interestingly, a strong difference in expression of Hebp1 and Glo1 was primarily observed in rapidly-progressing AD cases (death within 4 year period after diagnosis) (*Figure 3B and C*). These cases of AD are characterized by distinct pathological features and clinical parameters and are associated with a faster progression and more severe form of the disease (*Chitravas et al., 2011*; *Zafar et al., 2017a*).

We have additionally examined the publicly available mRNA expression datasets to determine the levels of Hebp1 in larger cohorts of AD patients (http://www.genenetwork.org/webqtl/main.py) (*Figure 3—figure supplement 1*). The datasets from Harvard Brain Tissue Resource Center (GN326, GN327, GN328) demonstrated significantly increased levels of *HEBP1* mRNA in prefrontal and primary visual cortex in AD patients stressing a strong relevance of Hebp1 to AD in humans. Taken together, these results support our findings that Hebp1 is indeed a novel protein dysregulated in Alzheimer's disease that is particularly associated with severe AD cases.

## Hebp1 is a neuronal protein upregulated in the brain of 3×Tg-AD mouse

To better understand the function of Hebp1 and how it contributes to the disease, we first examined its distribution in the brain. Immunoblot analyses of four brain areas in 12-month-old mice indicated that Hebp1 was most abundant in the hippocampus, followed by the brain stem and cortical areas

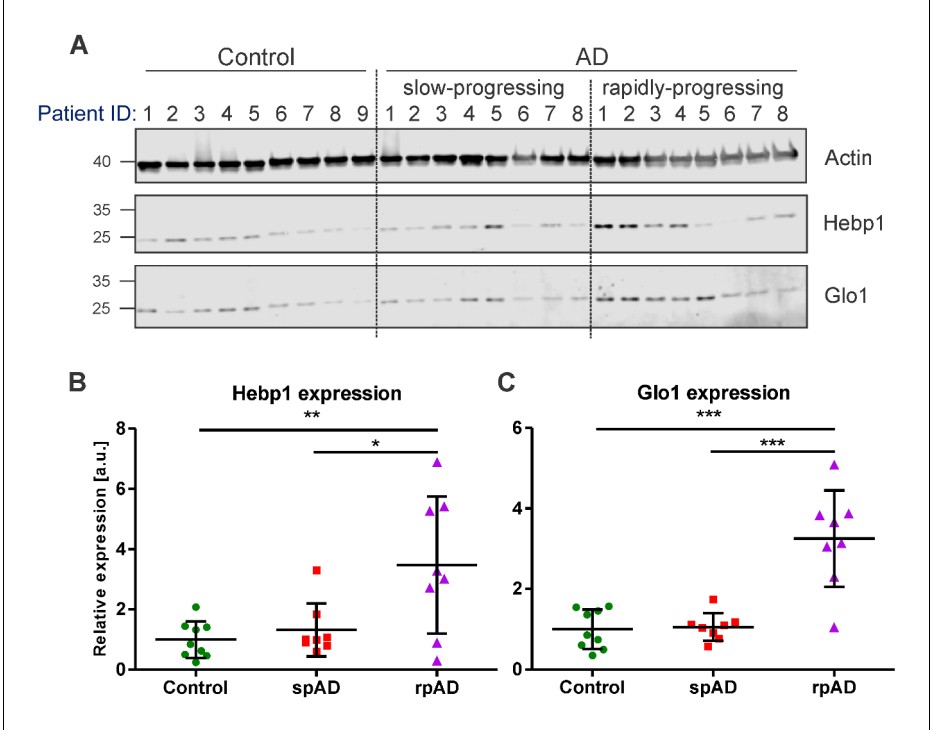

**Figure 3.** Hebp1 and Glo1 exhibit increased expression in brains of patients with rapidly-progressing forms of AD. (**A**) Immunoblot analysis of Hebp1 and Glo1 expression in slow-progressing (spAD) and rapidly-progressing (rpAD) AD cases and age-matched controls. Samples from nine control, eight slow-progressing AD and eight rapidly-progressing AD patients were used in this study. Detailed information on the patients is presented in *Table 2*. Quantification of (**B**) Hebp1 and (**C**) Glo1 levels in human samples. Error bars in graphs represent mean ± SD. Statistical significance in the datasets was assessed by one-way ANOVA followed by Bonferroni's multiple comparisons test for individual pairs of samples ($\alpha$ = 0.05): *p<0.025, **p<0.01, and ***p<0.0001.
DOI: https://doi.org/10.7554/eLife.47498.009
The following figure supplement is available for figure 3:

**Figure supplement 1.** *HEBP1* expression in publicly available transcriptome databases of Alzheimer's disease.
DOI: https://doi.org/10.7554/eLife.47498.010

---

(*Figure 4A*). Hebp1 was not detectable in the cerebellum. Significantly, Hebp1 became dramatically elevated in all four brain regions in age matched 3×Tg-AD mice (*Figure 4A*). Immunohistochemical analysis confirmed upregulated expression of Hebp1 in neocortex and hippocampus of 3×Tg-AD mice compared with wild-type controls (*Figure 4B*, *Figure 4—figure supplement 1*).

To identify which cell types express Hebp1 in the brain, we performed co-immunostaining of Hebp1 with cell-lineage specific markers (*Figure 4C*, *Figure 4—figure supplement 2*). Hebp1 is strongly expressed in Ctip2-immunoreactive neurons but is poorly associated with GFAP-stained astrocytes or Iba-1-labeled microglia in the hippocampus.

## Hebp1 interacts with mitochondrial contact site complex

While the role of Hebp1 in neurons has not been characterized, previous studies of Hebp1 and its homologues have left some clues regarding its potential function. Hebp1 has been proposed to participate in transport of heme from mitochondria to cytosol (*Jacob Blackmon et al., 2002*; *Taketani et al., 1998*). Furthermore, heme-binding protein 2/SOUL, a homologue of Hebp1, is involved in mediation of cell death by recruitment to mitochondria permeability transition pore (*Szigeti et al., 2006*; *Szigeti et al., 2010*). Given the heme-binding properties of Hebp1 and evolutionary similarity to Hebp2/SOUL, we hypothesized that it can perform one of these functions (*Fortunato et al., 2016*).

**Table 2.** Information of patients included in this study.

| Patient ID | Gender | Age | Disease duration (years) | Braak stages (AD) | Postmortem delays [hours] |
|---|---|---|---|---|---|
| Cont. 1 | Male | 86 | - | II/A | 06:45 |
| Cont. 2 | Male | 61 | - | I/0 | 03:03 |
| Cont. 3 | Male | 74 | - | II/A | 11:00 |
| Cont. 4 | Male | 86 | - | II/A | 06:45 |
| Cont. 5 | Female | 73 | - | I/0 | 04:03 |
| Cont. 6 | Male | 69 | - | II/A | 05:03 |
| Cont. 7 | Male | 68 | - | I/0 | 05:03 |
| Cont. 8 | Female | 64 | - | I/0 | 09:00 |
| Cont. 9 | Male | 67 | - | I/0 | 05:03 |
| spAD1 | Female | 72 | >4 | V/C | 09:30 |
| spAD2 | Female | 75 | >4 | V/C | 04:15 |
| spAD3 | Male | 78 | >4 | V/C | 09:30 |
| spAD4 | Male | 83 | <4 | V/C | 08:20 |
| spAD5 | Female | 56 | >4 | V/C | 07:00 |
| spAD6 | Male | 83 | >4 | III/0 | 07:25 |
| spAD7 | Female | 90 | >4 | IV/A | 09:55 |
| spAD8 | Female | 93 | >4 | V/C | 03:00 |
| rpAD1 | Male | 78 | <4 | V/C | 03:30 |
| rpAD2 | Female | 79 | <4 | V | 05:30 |
| rpAD3 | Female | 81 | <4 | III/B | 06:00 |
| rpAD4 | Male | 83 | <4 | VI/C | 05:30 |
| rpAD5 | Male | 83 | <4 | V/C | 08:20 |
| rpAD6 | Male | 70 | <4 | VI/C | 11:30 |
| rpAD7 | Male | 76 | <4 | VI/C | 06:30 |
| rpAD8 | Female | 77 | <4 | IV/A | 12:00 |

DOI: https://doi.org/10.7554/eLife.47498.011

In this case, mitochondrial or perimitochondrial localization of Hebp1 would be expected. Previous studies aiming to define the mitochondrial proteome led to ambiguous results with regard to mitochondrial localization of Hebp1 (*Calvo et al., 2016*; *Hung et al., 2014*; *Hung et al., 2017*). Subcellular fractionation of the mouse brain indicated that Hebp1 is present in both synaptosomal (P2) and crude mitochondrial (Mt) fractions where mitochondria are expected to be present (*Figure 5A*). We further demonstrated that the protein can be mitochondrially-associated by detecting the presence of Hebp1 from mitochondria isolated from cultured hippocampal neurons (*Figure 5B*). Supporting the biochemical data, we observed that a portion of EGFP-tagged Hebp1 expressed in rat primary neurons appears to be closely juxtaposed to mitochondria (visualized using Mitotracker) (*Figure 5C* and line scans in *Figure 5E and F*). In comparison to this, the signal from EGFP alone exhibited no correlated association with Mitotracker (*Figure 5C and D*).

We further examined the role of Hebp1 in neurons by identification of its binding partners. To this end, we performed immunoprecipitation of expressed EGFP-tagged Hebp1 in neurons using GFP-trap and investigated the co-precipitated proteins by MS analysis. The volcano plot represents the relative enrichment of detected proteins in either Hebp1-EGFP or EGFP (negative control) pull-downs (*Figure 6A*). Interestingly, the core components of mitochondrial contact site complex (MICOS), Mic60, Mic19 and Mic25, as well as proteins of outer mitochondria membrane associated with MICOS complex, SAMM50 and Mtx2, were enriched in Hebp1 Co-IP samples. We further

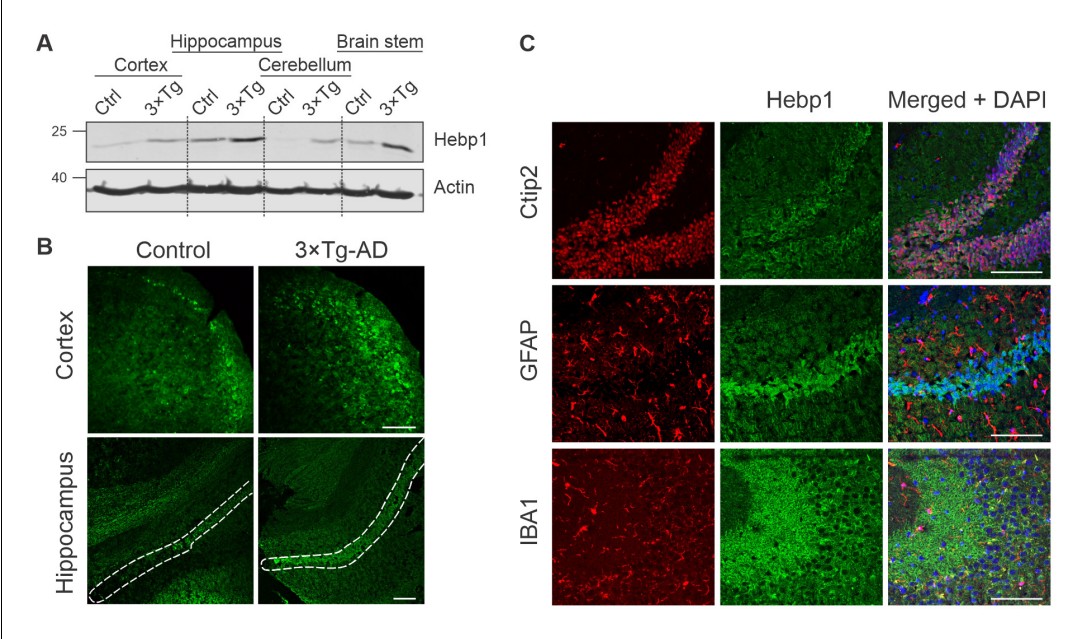

**Figure 4.** Analysis of Hebp1 expression in the brain of 3×Tg-AD mice. (**A**) Expression of Hebp1 in 12-month-old control and 3×Tg-AD mice by brain region. (**B**) Hebp1 immunostaining of the fronto-temporal cortex depicting primary motor and somatosensory areas and hippocampus (coronal sections). CA1 region is marked with the white dashed line. (**C**) Co-staining of Hebp1 with markers of CA1 and dentate gyrus neurons (Ctip2), astrocytes (GFAP) and microglia (IBA-1) in the hippocampus of 3×Tg-AD mice. Hepb1 is expressed predominantly in Ctip2-positive cells of hippocampus (neurons). All images were acquired from 12-month-old control or 3×Tg-AD mice. Scale bar is 100 μm. All data shown are representative of results obtained from three independent experiments.

DOI: https://doi.org/10.7554/eLife.47498.012

The following figure supplements are available for figure 4:

**Figure supplement 1.** Expression of Hebp1 in brains of control and 3×Tg-AD mice.

DOI: https://doi.org/10.7554/eLife.47498.013

**Figure supplement 2.** Expression of Hebp1 is localized to neurons.

DOI: https://doi.org/10.7554/eLife.47498.014

---

confirmed the interaction between Hebp1 with Mic60 by probing the immunoprecipitated complexes with their respective antibodies via immunoblotting (*Figure 6B*). Taken together these data indicate that Hebp1 locates in close proximity to the mitochondrial outer membrane where it interacts with MICOS complex potentially through association with outer mitochondria membrane proteins such as SAMM50 or Mtx2. In line with our observations, a very similar pattern of subcellular localization was observed in cells that were transfected with full-length Hebp2/SOUL (*Szigeti et al., 2006*), further hinting towards potential role of Hebp1 in the regulation of cell death.

## Hebp1 facilitates heme-mediated cytotoxicity

Heme metabolism, cell death response and AD are tightly interconnected. Dysregulation of proteins linked to heme metabolism has been reported in AD (Hani Atamna & Frey, 2004; *Schipper et al., 1995*). Moreover, Aβ can form a complex with heme which possesses strong peroxidase and superoxide activities that can contribute to oxidative stress and cytotoxicity during AD (*Atamna and Boyle, 2006*; *Chiziane et al., 2018*; *Ghosh et al., 2015*). Accumulation of Aβ around brain vasculature results in cerebral amyloid angiopathy (CAA), microvessel destruction and leakage of free heme into brain tissue (*Chiziane et al., 2018*). Due to its strong hydrophobicity, heme is almost exclusively bound to carrier proteins within cells. High concentrations of free heme are toxic to multiple cell types (*Gemelli et al., 2014*; *Owen et al., 2016*). We wondered whether Hebp1 could protect neurons by acting as an intracellular heme buffer to maintain the latter at low levels. To test this hypothesis, we successfully eliminated expression of Hebp1 in rat hippocampal neurons using CRISPR/Cas9 using three different gRNA sequences (*Figure 7A*, KO1-3). We then exposed both control and

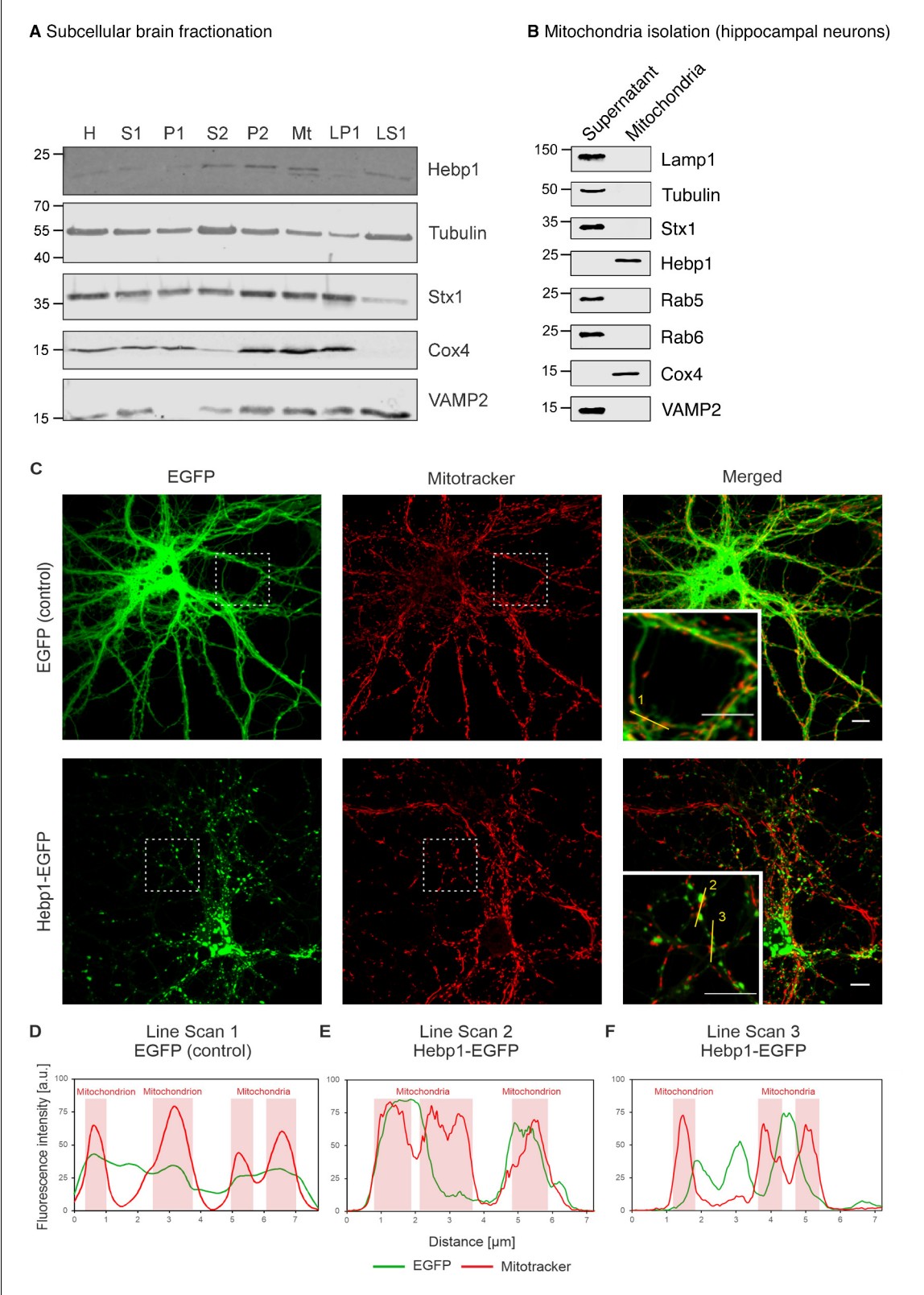

**Figure 5.** Hebp1 demonstrates perimitochondrial localization in neurons. (**A**) Brain fractionation was performed as described before (*Huttner et al., 1983*). Hebp1 was identified in crude mitochondria fraction (Mt). Fraction annotation: H – homogenate, S1 – supernatant 1, P1 – pellet 1, S2 – supernatant 2 (fraction of soluble proteins), P2 – pellet 2 (synaptosomes), Mt – mitochondria, LP1 – lysate pellet 1 (plasma membrane fraction of synaptosomes), LS1 – lysate supernatant 1 (soluble fraction of synaptosomes). 20 µg of each fraction were loaded on the gel, except for LS1 (6 µg). (**B**)
*Figure 5 continued on next page*

*Figure 5 continued*

Mitochondria were isolated from cultured hippocampal neurons as described previously (*Wieckowski et al., 2009*). Hebp1 was only detected in isolated mitochondria together with Cox4 while markers for endo-lysosomal, synaptic and plasma membrane compartments (Lamp1, tubulin, Stx1, Rab5, Rab6 and VAMP2) were exclusively present in the supernatant. (**C**) Localization analysis of mitochondria (mitotracker), Hebp1-EGFP and EGFP alone in cultured rat hippocampal neurons (DIV14). Hebp1 puncta is associated with mitochondria. Representative line scans (golden lines in the inserts; location of the numbers correspond to the starting point of each analysis) were traced for EGFP control (**D**) and Hebp1-EGFP (**E–F**). Line scan analyses indicate that at least some of the Hebp1-EGFP puncta appear to be contacting mitochondria (**E–F**). Scale bar is 10 μm. All data shown are representative of results obtained from three independent experiments.
DOI: https://doi.org/10.7554/eLife.47498.015

Hebp1-deficient neurons to exogenous hemin and measured the extent of cytotoxicity by determining the activity of the dead cell protease.

Consistent with the cytotoxic effects of high heme levels, treatment of wildtype (WT) and control neurons (LUC) with hemin resulted in 2.69 (±0.15) and 2.86 (±0.23) fold increase in cell death for WT and LUC respectively as compared to exposure to vehicle only (*Figure 7B*). Strikingly, Hebp1-deficient neurons exhibited no significant cell death. This effect was consistently observed in both the gRNA sequences used (*Figure 7B*, KO1 and KO2). Thus, contrary to our expectations, Hebp1 does not protect neurons from excessive heme. Rather, the protein apparently mediates its toxic effect. To confirm the specificity of heme as the inducer of Hebp1-dependent cell death, we also exposed the neurons to *tert*-butyl-hydroperoxide (stable analog of hydrogen peroxide) or staurosporine, both of which are known inducers of apoptotic cell death (*Belmokhtar et al., 2001*; *Zhao et al., 2017*). As expected, all 3 groups of neurons exhibited increased levels of cellular toxicity, confirming that loss of Hebp1 is protective towards hemin-induced cytotoxicity (*Figure 7C and D*). We also evaluated if over-expression of Hebp1 could affect cytotoxicity. Increased hemin-induced cytotoxicity was observed for Hebp1-EGFP expressing neurons as compared to EGFP-expressing neurons (*Figure 7—figure supplement 1*). However, this increase did not reach statistical significance.

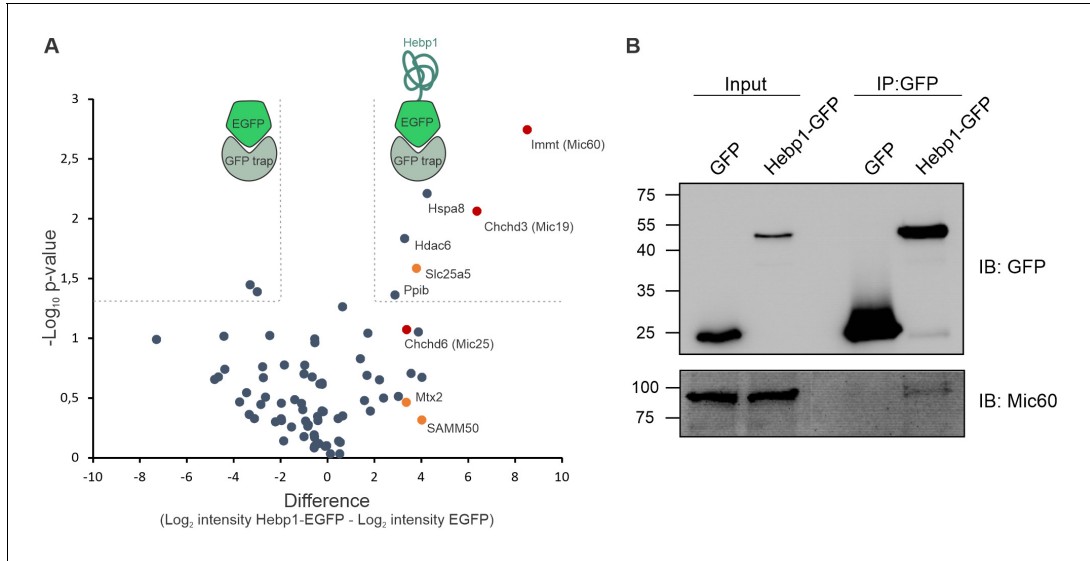

**Figure 6.** Hebp1 interactome reveals its association with mitochondrial contact site complex (MICOS). (**A**) Hebp1 interactome obtained by mass spectrometry analysis of proteins co-immunoprecipitated from primary cortical neurons with Hebp1-EGFP or EGFP (negative control). Enrichment of mitochondria contacts site complex (MICOS) proteins (red) or MICOS-associated proteins (orange). Dashed line represents a cut-off for significantly different proteins between Hebp1-EGFP and control pulldown with at least 4-fold change. (**B**) Validation of Hebp1-Mic60 interaction by immunoblotting. All data shown are representative of results obtained from three independent experiments.
DOI: https://doi.org/10.7554/eLife.47498.016
The following source data is available for figure 6:

**Source data 1.** Hebp1 interactome.
DOI: https://doi.org/10.7554/eLife.47498.017

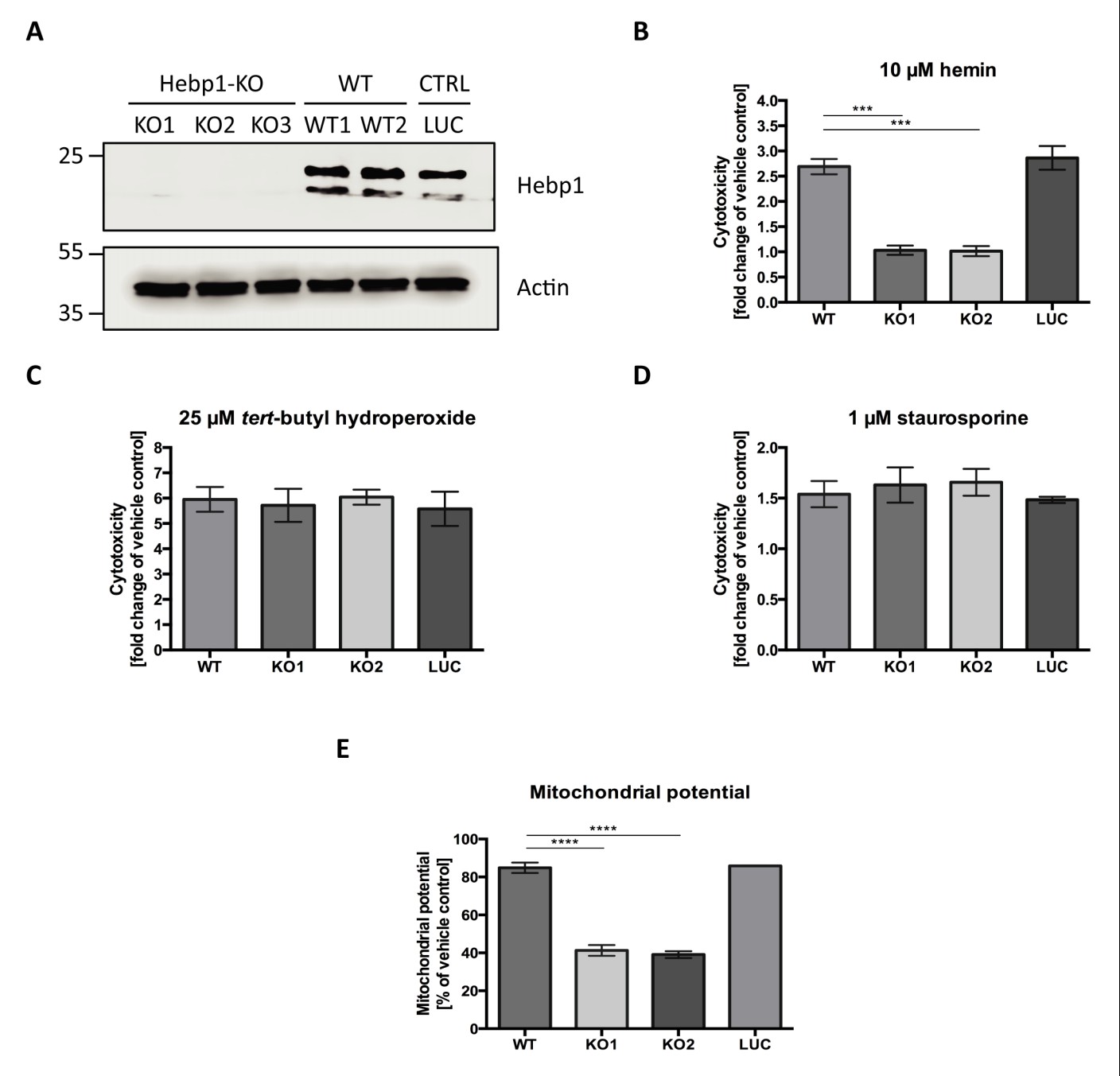

**Figure 7.** Hebp1 mediates neurotoxicity upon heme overload. (A) Knockout of Hebp1 in neurons by CRISPR/Cas9. (B) Measurement of cytotoxicity using the MultiTox-Glo reagent (Promega) was performed 24 hr after stimulation with 10 µM hemin or vehicle. Hebp1-deficient neurons are resistant to heme-mediated cytotoxicity. Wildtype, control and Hebp1-deficient neurons demonstrate similar elevated cytotoxicity in response to 3 hr treatment with 25 µM *tert*-butyl hydroperoxide (C) and 1 µM staurosporine (D). (E) Hemin treatment induces significantly higher reduction of mitochondrial potential in Hebp1-deficient neurons in comparison to wildtype and control neurons. Mitochondrial potential was measured using the Mitochondrial Membrane Potential Assay kit (Cell Signaling). All bar charts represent mean ± SEM. Statistical significance in the datasets was assessed by one-way ANOVA followed by Student's t-test comparison for individual pairs of samples: ***$p < 0.005$ and ****$p < 0.001$. All data shown are representative of results obtained from three independent experiments.

DOI: https://doi.org/10.7554/eLife.47498.018

The following figure supplement is available for figure 7:

**Figure supplement 1.** Cytotoxicity is influenced by hemin concentrations rather than cellular levels of Hebp1.
DOI: https://doi.org/10.7554/eLife.47498.019

Instead, statistically significant increases in cytotoxicity were observed when hemin concentrations were raised from 5 to 10 µM within the EGFP or EGFP-Hebp1 expressing group. These results indicate that endogenous levels of Hebp1 are sufficient to trigger neuronal cell death that is, in turn, influenced by heme concentrations.

As described in the preceding paragraphs, Hebp1 is localized to mitochondria via its interaction with the MICOS complex. Of note, Mic60 plays important roles in critical aspects of mitochondrial function and integrity and has been linked to mitochondrial-associated apoptosis (*Van Laar et al., 2018*). In particular, reduction of Mic60 in cells has been associated with release of Cytochrome C (Cyt$_C$) and increased sensitivity to apoptosis triggers (*Yang et al., 2012*). Release of Cyt$_C$, often associated with loss of mitochondrial potential, is followed by Apaf1 activation and the ensuing conversion of procaspase 9 into caspase 9 and procaspase 3 into caspase 3, thereby committing the cells into apoptosis (*Franklin and Robertson, 2007*; *Zou et al., 1999*). The interaction of Hebp1 with Mic60 suggests that Hebp1-dependent apoptosis in response to excess heme might be initiated at the mitochondria. Indeed, exposure of neurons to hemin decreases mitochondrial membrane potential (MMP) in wildtype, control as well as Hebp1-deficient neurons (*Figure 7E*). However, a steeper decrease in MMP was observed in Hebp1-deficient neurons (58.7 ± 2.83% and 60.9 ± 1.77% in KO1 and KO2 neurons) as compared to the other two groups (15.1 ± 2.77% and 14.1 ± 0.22% in wildtype and control neurons, respectively) strongly suggesting that effects on mitochondrial membrane potential alone cannot be the sole explanation for Hebp1's role in causing neuronal apoptosis.

Indeed, using mitochondria isolated from these neurons, we observe that Cyt$_C$ was released from these organelles in all neurons (wildtype, control and Hebp1-deficient) treated with hemin (*Figure 8A and B*), confirming that this step of mitochondria-associated apoptosis is not affected by loss of Hebp1 function. Strikingly, we observe that while procaspase 9 was cleaved in wildtype and control neurons upon hemin-treatment, the protein was not activated in Hebp1-deficient neurons (*Figure 8C*). In agreement with this, hemin exposure caused a dramatic increase in the population of caspase 3$^+$/7$^+$ cells in wildtype (58.5 ± 2.89%) and control neurons (62.4 ± 1.88%) but not in Hebp1-deficient neurons (8.69 ± 1.55% and 9.86 ± 2.09% in KO1 and KO2, respectively) (*Figure 8D and E*). Noteworthy, Hebp1, found almost exclusively in the mitochondrial fraction in untreated wildtype and control neurons, was released from the mitochondria of these neurons upon exposure to hemin (*Figure 8B*). Collectively, these results indicate that release of mitochondrial Hebp1 in the presence of excessive heme is a critical trigger linking mitochondrial damage to neuronal apoptosis via activation of the procaspase 9 pathway. Interestingly, Mic60 was also released from mitochondria, but only in Hebp1-deficient cells, while Cox4 remained in the mitochondrial pellet for all groups of neurons. The significance of this remains unclear.

We also examined if absence of Hebp1 could also protect neurons from exposure to Aβ$_{42}$. As expected, cytotoxicity was observed in both wildtype (40.2 ± 1.23%) and control neurons (42.8 ± 1.15%) treated with Aβ$_{42}$ (*Figure 9*). Concurrent exposure of these neurons to Aβ$_{42}$ and hemin caused an additional 2-fold increase in cytotoxicity (87.9 ± 1.04% for wildtype and 89.1 ± 0.63% for control neurons), an additive effect presumably caused by formation of an Aβ-heme complex that can contribute to oxidative stress and cytotoxicity during AD (*Atamna and Boyle, 2006*; *Chiziane et al., 2018*; *Ghosh et al., 2015*). In contrast to this, very low levels of cytotoxicity were detected when Hebp1-KO neurons were exposed to Aβ$_{42}$ alone (7.51 ± 0.70% and 7.29 ± 0.96% in KO1 and KO2, respectively), similar to hemin treatment (9.65 ± 0.77% and 9.42 ± 0.87% in KO1 and KO2, respectively). Even more significantly, cytotoxicity levels remained low even after concurrent treatment with Aβ$_{42}$ and hemin (22.9 ± 0.84% and 23.9 ± 0.84% in KO1 and KO2, respectively). Thus, Hebp1 participates in converging pathways triggered by Aβ$_{42}$ and heme that cause cytotoxicity via apoptotic cell death in neurons.

## Discussion

In this study, we identified several potential presymptomatic brain markers of AD by examining changes in brain proteome between wild-type and 3×Tg-AD mice. Of these, Hebp1 is consistently elevated in the brains of 3×Tg-AD mice from early stage of the disease and is also significantly increased in postmortem brains of patients affected by rapidly-progressing forms of AD. Hebp1 appears to be mainly expressed in neurons where it is associated with mitochondria via the MICOS

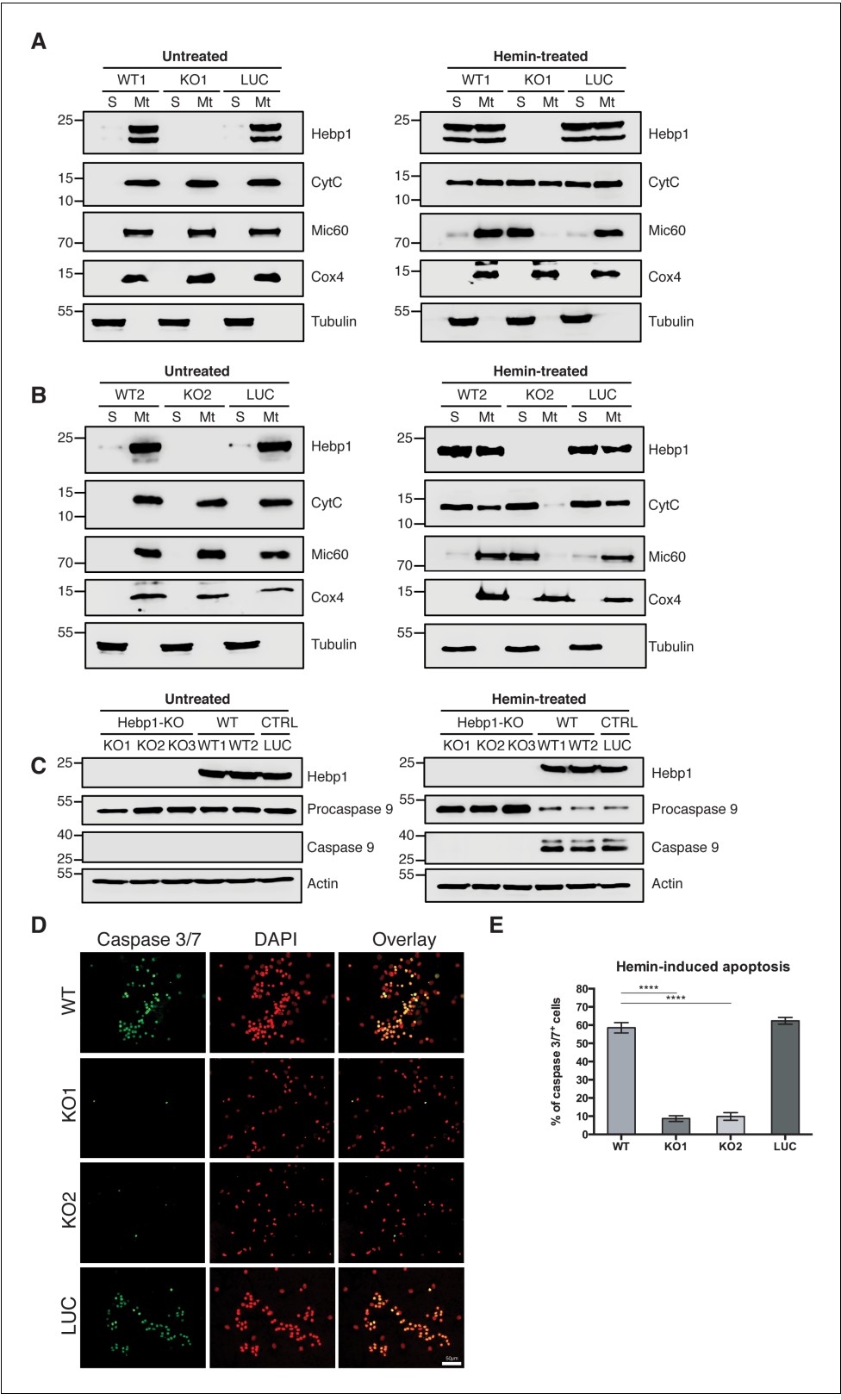

**Figure 8.** Neuronal cell death occurs by triggering mitochondrial-dependent apoptotic pathway in Hebp1-expressing neurons. Western blot analyses of (**A and B**) $Cyt_C$ and Mic60 leakages and (**C**) caspase 9 activation in Hebp1-deficient, wildtype and control neurons. Wildtype and control neurons exhibited high levels of activated caspase 9 concomitant with mitochondrial release of $Cyt_C$ and Mic60 into the cytosol (**S**). Hebp1 release was also coupled with leakage of $Cyt_C$ and Mic60 in these cells. In contrast, Hebp1-deficient neurons displayed no apparent activation of caspase 9 despite

*Figure 8 continued on next page*

*Figure 8 continued*

leakages of Cyt$_C$ and Mic60 from neuronal mitochondria (Mt). (**D**) Wildtype, control and Hebp1-deficient neurons were treated with 10 μM hemin for 24 hr. Apoptotic cells were visualized by fluorescence staining corresponding to caspase 3/7 activation (see Materials and methods). Hebp1-deficient neurons demonstrated resistance to apoptosis upon heme overload, whereas wildtype and control neurons exhibited high levels of caspase 3/7 activity. (**E**) Quantification of the data represented by the images shown in (**D**). All bar charts represent mean ± SEM. Statistical significance in the datasets was assessed by one-way ANOVA followed by Student's t-test comparison for individual pairs of samples: ****p<0.001. All data shown are representative of results obtained from three independent experiments.

DOI: https://doi.org/10.7554/eLife.47498.020

complex. Strikingly, knockdown of Hebp1 expression in neurons protects them from both heme- and Aβ42-induced apoptosis, suggesting that Hebp1 plays a role in sensitizing neurons to cytotoxicity over the course of AD progression.

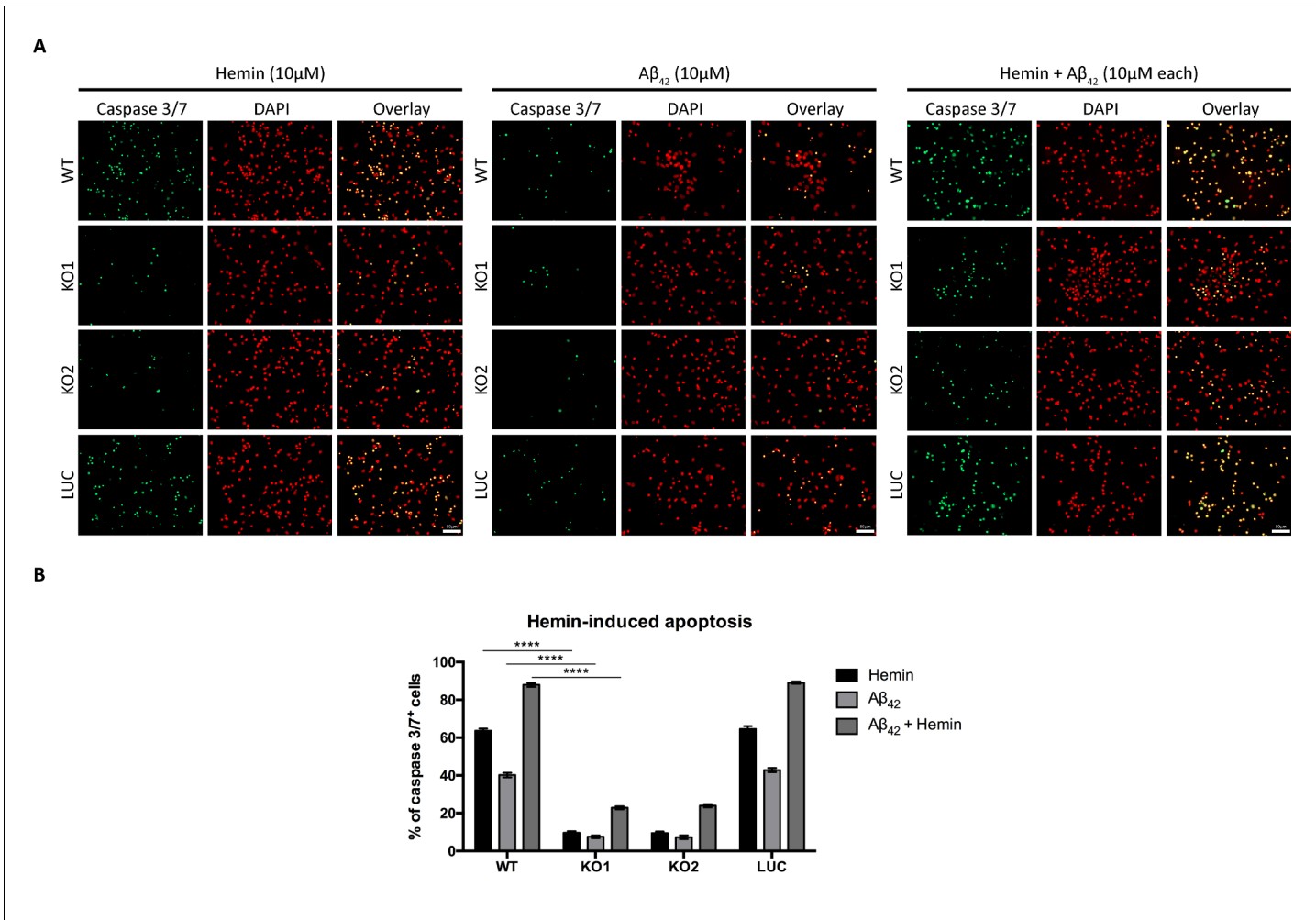

**Figure 9.** Knockout of Hebp1 in neurons is neuroprotective against hemin and/or Aβ$_{42}$-induced neuronal cell death. (**A**) Wildtype, control and Hebp1-deficient neurons were treated with 10 μM hemin or 10 μM Aβ$_{42}$ oligomers or both simultaneously (10 μM each). Apoptotic cells were observed for caspase 3/7 activation via fluorescence staining (see Materials and methods). Hebp1-deficient neurons showcased resistance to apoptosis upon heme and/or Aβ$_{42}$ overload, with wildtype and control neurons exhibiting opposing effects of highly elevated caspase 3/7 activity. (**B**) Quantification of the data represented by the images shown in (**A**). All bar charts represent mean ± SEM. Statistical significance in the datasets was assessed by two-way ANOVA followed with Bonferroni corrections for individual pairs of samples: ****p<0.001. All data shown are representative of results obtained from three independent experiments.

DOI: https://doi.org/10.7554/eLife.47498.021

Our study indicated that proteins related to cell death, mitochondria function and seizures are among the first affected in AD. Alterations of cytoskeleton-related proteins became apparent only at the late stage of the disorder which coincides with the timeline of tau aggregation. Several proteins in our dataset exhibited gradual increases in expression that correlated with disease progression in 3×Tg-AD mice. These included proteins that are involved in mRNA processing such as Matrin-3 and Nono that form a complex involved in DNA damage response and recognition and retention of incorrectly processed mRNA in nucleus (*Salton et al., 2010*; *Zhang and Carmichael, 2001*). Similar expression patterns were detected for Hnrnpm and Hnrnpl, which regulate alternative splicing (*Fei et al., 2017*; *Passacantilli et al., 2017*). Bai and colleagues have previously found U1 snRNP components to be enriched in insoluble brain proteome of AD patients and demonstrated impaired splicing of AD-related transcripts (*Bai et al., 2013*). While the role of mRNA processing and alternative splicing in AD has not been studied extensively, our observations point towards dysregulation of this process in 3×Tg-AD mice. We also observed that several inhibitors of serine proteases (serpins) appear to progressively decrease in expression during the monitoring period. Interestingly, a recent study of the hippocampal proteome in the 5 × FAD mouse model of AD also identified downregulation of serpins (*Gurel et al., 2018*). Protease inhibitors hold a potential as therapeutic targets since their levels in the brain can be restored through injection of recombinant protein. This approach was recently tested with metalloprotease inhibitor TIMP2 which was injected intraperitoneally to reach the brain and improve the cognitive function of aged mice (*Castellano et al., 2017*).

The present study uncovered two proteins, Hebp1 and Glo1, that were highly elevated already at the presymptomatic stage in 3×Tg-AD mice, suggesting that they could be of potential relevance as early AD markers. Both proteins were also expressed at significantly higher levels - particularly in rapidly progressing cases of AD - indicating that they could be of use to identify this group of patients. While Glo1 has been previously linked to neurodegeneration (*Chen et al., 2004*; *More et al., 2013*), our study is the first to report the involvement of Hebp1 in AD.

According to our data, Hebp1 is predominantly expressed in neurons in both wild-type and 3×Tg-AD mice where it is associated with mitochondria via interaction with the MICOS complex. Interestingly, Mic60, a component of the MICOS complex that binds Hebp1, is also an important player in cell death. Loss of Mic60 increases the rate of apoptosis due to dissipation of cristae junctions and intensifies leakage of $Cyt_C$ from mitochondria to cytosol (*Yang et al., 2012*). Moreover, SOUL, a homolog of Hebp1, promotes cell death presumably through permeabilization of mitochondria membranes (*Szigeti et al., 2006*; *Szigeti et al., 2010*). Together, these findings suggest that Hebp1 itself could also be involved in causing cellular toxicity.

What implications might heme-mediated cell death have for AD? Loss of neurons is a key event leading to cognitive decline in AD and is mainly attributed to intensified apoptosis (*Cotman and Su, 1996*). Overexpression of proteins counteracting apoptotic response has been previously shown to decrease pathology in 3×Tg-AD mice (*Rohn et al., 2008*). Degradation of heme by heme oxygenase-1 was demonstrated to reduce cytotoxicity caused by Aβ1–42 peptide (*Hettiarachchi et al., 2014*). Heme synthesis is impaired in AD and accumulation of immature heme species can be a potential source of heme overload (Hani Atamna & Frey, 2004). Excessive heme can also come from circulating blood. Cerebral amyloid angiopathy contributes significantly to AD both in human and the 3×Tg-AD model starting from the early stage of the disorder (*Ghiso et al., 2010*; *Li and Praticò, 2015*). Moreover, pathophysiological changes in brain vasculature have also been reported in 3×Tg-AD mice (*Grammas et al., 2014*; *Lin et al., 2014*). Thus, CAA may lead to disruption of brain vasculature and release of heme outside the vessels (*Chiziane et al., 2018*; *Natté et al., 2001*) and is also associated with apoptosis (*Fossati et al., 2010*; *Mattson, 2000*). Collectively, these data highlight a possible link between impairments of heme metabolism, neuronal loss and increased expression of Hebp1 early in AD.

In agreement with this, we observe that Hebp1 is intimately involved in heme-induced neuronal death. Indeed, while neurons expressing Hebp1 showed dramatically elevated levels of cell death, Hebp1-deficient neurons were resistant to hemin-induced apoptosis. Likewise, neuronal cell death via apoptosis triggered upon exposure to $Aβ_{42}$ was also significantly attenuated in Hebp1-deficient neurons. Remarkably, Hebp1 sits at a critical, albeit currently unknown, position between upstream activating events (mitochondrial membrane potential changes and $Cyt_C$ leakage) and the initiation of the caspase cascade (*Galluzzi et al., 2018*). In Hebp1-deficient neurons, loss of Hebp1 and the

consequent lack of release of the protein into the cytosol halted the activation of caspases 9 and 3/7 in spite of the occurrence of upstream activating events, suggesting that it could play a role in regulating apoptosome formation which is vital in cleaving procaspase 9 to its functional form (*Saleh et al., 1999*; *Zou et al., 1999*). The significance accompanying the observed concurrent release of Mic60 in Hebp1-deficient neurons observed in response to hemin-treatment remains unclear despite reduced levels of Mic60 being linked to $Cyt_C$ release (*Yang et al., 2012*). This would be an important subject for further studies.

Hebp1 might also play further roles in AD pathology in addition to its newly uncovered functions in this study. Two previous publications demonstrated that N-terminal cleavage of Hebp1 by cathepsin D results in generation of 21 amino acid long peptide called F2L that is capable of binding FPRL1/FPR2 receptor on the surface of mouse neutrophils and promote their migration (*Devosse et al., 2011*; *Gao et al., 2007*). In the mouse brain, FPR2 is expressed predominantly by activated microglia (*Cui et al., 2002*). Moreover, FPRL1-positive microglia was shown to be recruited to Aβ plaques in AD patients (*Le et al., 2001*). In our dataset, expression of the Hebp1 protease cathepsin D also strongly correlates with aging which indicates the possibility of the progressive F2L accumulation in 3×Tg-AD mice with age. Increased expression of cathepsin D in hippocampus of AD patients was also reported previously (*Hondius et al., 2016*). Thus, it is possible that cleavage of Hebp1 by cathepsin D in neurons may additionally generate the soluble F2L peptide to recruit activated microglia and modulate inflammatory response during AD.

Overall, our results provide a quantitative proteome map of AD progression in the 3×Tg-AD transgenic mouse model and identify several novel protein candidates that could serve as putative presymptomatic markers of the disease. These data can serve as a starting point to allow for a more thorough investigation of these markers in relation to their roles in AD pathogenesis.

## Materials and methods

**Key resources table**

| Reagent type (species) or resource | Designation | Source or reference | Identifiers | Additional information |
|---|---|---|---|---|
| Gene (*Homo-sapiens*) | *HEBP1* | Origene Gene ID: 50865 | Cat#: RC201873 | Complete CDS sequence was used in this study |
| Strain, strain background (*M. musculus*) | 3 × Tg AD mice (B6.129.*Thy* tr.tg-/-) | PMID:12895417 | | Provided by Prof. Wolfgang Härtig |
| Strain, strain background (*M. musculus*) | B6;129 (129/sv C57bl6 WT) | PMID:12895417 | | Provided by Prof. Wolfgang Härtig |
| Biological sample (*R. norvegicus*) | Primary cortical neurons | InVivos, Singapore | | Freshly isolated from postnatal Day 0 *Rattus norvegicus* pups |
| Biological sample (*R. norvegicus*) | Primary hippocampal neurons | InVivos, Singapore | | Freshly isolated from postnatal Day 0 *Rattus norvegicus* pups |
| Antibody | anti-Hebp1 (Rabbit, polyclonal) | Invitrogen | Cat#: PA5-30609 RRID: AB_2548083 | WB (1:1000), IHC (1:100) |
| Antibody | anti-Glyoxalase 1 (Mouse, monoclonal) | GeneTex | Cat#: GTX628890 RRID: AB_2787101 | WB (1:1000) |
| Antibody | anti-carbonic anhydrase I (Rabbit, polyclonal) | Novus Biologicals | Cat#: NBP1-88191 RRID: AB_11017594 | WB (1:250) |
| Antibody | anti-α-tubulin (Mouse, monoclonal) | Synaptic Systems | Cat#: 302 211 RRID: AB_887862 | WB (1:5000) |

*Continued on next page*

*Continued*

| Reagent type (species) or resource | Designation | Source or reference | Identifiers | Additional information |
|---|---|---|---|---|
| Antibody | anti-β-actin (Rabbit, polyclonal) | Synaptic Systems | Cat#: 251 003 RRID: AB_11042458 | WB (1:5000) |
| Antibody | anti-GFP (Rabbit, polyclonal) | Synaptic Systems | Cat#: 132 002 RRID: AB_887725 | WB (1:5000) |
| Antibody | anti-Rab5 (Mouse, monoclonal) | Synaptic Systems | Cat#: 108 111 RRID: AB_2619777 | WB (1:1000) |
| Antibody | anti-Rab6 (Rabbit, polyclonal) | Synaptic Systems | Cat#: 273 003 RRID: AB_2619999 | WB (1:1000) |
| Antibody | anti-Lamp1 (Rabbit, polyclonal) | Abcam | Cat#: ab24170 RRID: AB_775978 | WB (1:500) |
| Antibody | anti-Mic60 (Mouse, monoclonal) | Abcam | Cat#: ab110329 RRID: AB_10859613 | WB (1:1000) |
| Antibody | anti-Cox4 (Rabbit, polyclonal) | Synaptic Systems | Cat#: 298 002 RRID: AB_2620041 | WB (1:1000) |
| Antibody | anti-Cyt$_C$ (Rabbit, monoclonal) | Cell Signaling | Cat#: 11940S RRID: AB_2637071 | WB (1:1000) |
| Antibody | anti-caspase 9 (Rabbit, monoclonal) | Abcam | Cat#: ab185719 RRID: AB_1140716 | WB (1:1000) |
| Antibody | anti-Sodium Potassium ATPase, subunit α1 | Abcam | Cat#: ab7671 RRID: AB_306023 | WB (1:1000) |
| Antibody | anti-syntaxin 1 (Mouse, monoclonal) | Synaptic Systems | Cat#: 110 001 RRID: AB_887843 | WB (1:1000) |
| Antibody | anti-VAMP2 (Mouse, monoclonal) | Synaptic Systems | Cat#: 104 211 RRID: AB_887811 | WB (1:10000) |
| Antibody | anti-phospho-tau (Ser400; Thr403;Ser404) (Rabbit, polyclonal) | Cell Signaling | Cat#: 11837S Product discontinued | WB (1:1000) |
| Antibody | anti-Ctip2 (Rat monoclonal) | Abcam | Cat#: ab18465 RRID: AB_2064130 | IHC (1:100) |
| Antibody | anti-GFAP (Mouse, monoclonal) | Synaptic Systems | Cat#: 173 011 RRID: AB_2232308 | IHC (1:500) |
| Antibody | anti-GFAP (Mouse, monoclonal) | Sigma | Cat#: C9205 RRID: AB_476889 | IHC (1:250) |
| Antibody | anti-IBA-1 (Guinea pig, polyclonal) | Synaptic Systems | Cat#:234 004 RRID: AB_2493179 | IHC (1:100) |
| Antibody | anti-NeuN (Guinea pig, polyclonal) | Synaptic Systems | Cat#: 266 004 RRID: AB_2619988 | IHC (1:200) |
| Transfected construct | FUGW (plasmid) | David Baltimore's Lab (Caltech) | Addgene plasmid #14883 RRID:Addgene_14883 | 3rd gen lentiviral plasmid with hUbC-driven EGFP |
| Transfected construct | psPax2 | Didier Trono's Lab (EPFL) | Addgene plasmid #12260 RRID:Addgene_12260 | 2nd generation lentiviral packaging plasmid |
| Transfected construct | pCMV-VSV-G | Bob Weinberg's Lab (MIT) | Addgene plasmid #8454 RRID:Addgene_8454 | Envelope protein for producing lentiviral and MuLV retroviral particles. |
| Transfected construct | FUGW-Hebp1 (plasmid) | This paper | | 3rd gen lentiviral plasmid with hUbC-driven Hebp1-EGFP |

*Continued on next page*

*Continued*

| Reagent type (species) or resource | Designation | Source or reference | Identifiers | Additional information |
|---|---|---|---|---|
| Transfected construct | LentiCRISPRv2 | Feng Zhang's Lab (Broad Institute) | Addgene plasmid #52961 RRID:Addgene_52961 | Replaces original lentiCRISPRv1 (Addgene Plasmid 49535) and produces ~ 10 fold higher titer virus. 3rd generation lentiviral backbone |
| Transfected construct | pLenti-CRISPR-Hebp1-KD1 | This paper | | LentiCRISPRv2 with inserted sgRNA Hebp1-KO1 targeting rat Hebp1 |
| Transfected construct | pLenti-CRISPR-Hebp1-KD2 | This paper | | LentiCRISPRv2 with inserted sgRNA Hebp1-KO2 targeting rat Hebp1 |
| Transfected construct | pLenti-CRISPR-Hebp1-KD3 | This paper | | LentiCRISPRv2 with inserted ssgRNA Hebp1-KO3 targeting rat Hebp1 |
| Transfected construct | pLenti-CRISPR-Luc | This paper | | LentiCRISPRv2 with inserted ssgRNA Luc targeting Luciferase. Used as a negative control For knockout experiments |
| Sequenced-based reagent | sgRNA: Hebp1 (KO1) | This paper | | 5'-CCCAGC ATGGTGACGCCGTG-3' |
| Sequenced-based reagent | sgRNA: Hebp1 (KO2) | This paper | | 5'-TGGCAGGT TCTAAGCACCGG-3' |
| Sequenced-based reagent | sgRNA: Hebp1 (KO3) | This paper | | 5'-CCGGTGC TTAGAACCTGCCCA-3' |
| Sequenced-based reagent | sgRNA: Luciferase (Luc) | This paper | | 5'-TCATATT CGTTAAAGCCCGG-3' |
| Peptide, recombinant protein | trypsin | Promega | Cat. #: V5113 | |
| Peptide, recombinant protein | papain enzymatic solution | Worthington Biochemical Corporation | Cat. #: LS003126 | |
| Peptide, recombinant protein | DNaseI | Sigma | Cat. #: D5025 | |
| Peptide, recombinant protein | Aβ$_{42}$ | Abcam | Cat. #: ab120301 | final concentration: 10 μM |
| Commercial assay or kit | Pierce 660 nm Protein Assay | Pierce | Cat. #: 22660 | |
| Commercial assay or kit | MitoTracker Red CMXRos | Life Technologies | Cat. #: M5712 | final concentration: 10 nM |
| Commercial assay or kit | MultiTox-Glo reagent, G9270 | Promega | Cat. #: G9270 | |
| Commercial assay or kit | CellEvent Caspase-3/7 Green Detection Reagent | Sigma | Cat. #: C10723 | |

*Continued on next page*

*Continued*

| Reagent type (species) or resource | Designation | Source or reference | Identifiers | Additional information |
|---|---|---|---|---|
| Commercial assay or kit | Mitochondrial Membrane Potential Assay kit | Cell Signaling | Cat. #: 13296 | final concentration of TMRE dye: 200 nM |
| Chemical compound, drug | protease/phosphatase inhibitors | Pierce | Cat. #: 88669 | |
| Chemical compound, drug | RapiGest | Waters | Cat. #: 186002123 | |
| Chemical compound, drug | DTT | Thermo Fisher Scientific | Cat. #: 20290 | |
| Chemical compound, drug | chloroacetamide | Sigma | Cat. #: 22790 | |
| Chemical compound, drug | L-alanyl-L-glutamine | Millipore | Cat. #: K0302 | |
| Chemical compound, drug | MEM-Vitamine | Sigma | Cat. #: K0373 | |
| Chemical compound, drug | Mito+Serum extender | Corning Costar | Cat. #: 355006 | |
| Chemical compound, drug | FUDR | Sigma | Cat. #: F0503 | |
| Chemical compound, drug | Thioflavin S | Santa Cruz | Cat. #: CAS 1326-12-1 | |
| Chemical compound, drug | hemin | Sigma | Cat. #: 51289 | final concentration: 10 µM |
| Chemical compound, drug | *tert*-butyl-hydroperoxide | Sigma | Cat. #: 458139 | final concentration: 25 µM |
| Chemical compound, drug | 1 µM staurosporine | Santa Cruz | Cat. #: sc-3510 | final concentration: 1 µM |
| Software, algorithm | MaxQuant, software package version 1.5.0.25 | (*Cox and Mann, 2008*) | RRID:SCR_014485 | |
| Software, algorithm | Andromeda search engine | (*Cox et al., 2011*) | | |
| Software, algorithm | *Perseus, version 1.5.5.3* | (*Cox and Mann, 2008*) | RRID:SCR_015753 | |
| Software, algorithm | Ingenuity Pathway Analysis | QIAGEN Inc | RRID:SCR_008653 | |
| Software, algorithm | GraphPad Prism | GraphPad Prism (https://graphpad.com) | RRID:SCR_015807 | |
| Other | Vectashield mounting medium containing DAPI | Vector Laboratories | Cat. #: VEC-H-1500 RRID:AB_2336788 | |

## Mice

All animal procedures used in this study here fully comply with the guidelines as stipulated in the section 4 of the Animal Welfare Law of the Federal Republic of Germany (section 4 of TierSchG, Tierschutzgesetz der Bundesrepublik Deutschland). 3×Tg-AD mice (B6.129.*Thy* tr.tg-/-), generated on a mixed 129/sv-C57bl6 genetic background (*Oddo et al., 2003b*), and control B6;129 (129/sv C57bl6 WT) mice were used for the experiments. For preparation of primary neurons, Wistar rats originated from the local animal facility were used. All animals were maintained under 12L/12D cycle with food and water ad libitum.

## Sample collection for mass spectrometry

Whole brains of male mice (both control and 3×Tg-AD) were collected at 2, 6, 12 and 18 months of age in four biological replicates. Half of the brain was immersion-fixed with 4% phosphate-buffered paraformaldehyde and used for the immunohistochemical analysis. The other half was homogenized by a glass-Teflon homogenizer (RW20-DZM, IKA) in 3 ml ice-cold homogenization buffer (containing protease/phosphatase inhibitors, Pierce, Rockford, IL, USA, 88669) at 900 rpm for nine strokes. Thereafter, the homogenate was centrifuged for 2 min at $3000 \times g$, 4°C in S100AT4 rotor (SORVALL) to remove cell debris. Next, the supernatant was transferred to a new tube and additionally centrifuged for 12 min at $14500 \times g$ in S100AT4 rotor at 4°C to obtain the soluble fraction of brain proteins (supernatant).

## Mass spectrometry sample preparation and measurement

The protein concentration of the samples was measured by Pierce 660 nm Protein Assay according to the manufacturer's protocol. 40 μg of protein were used for proteomic analysis. Proteins were precipitated with four volumes of ice-cold acetone overnight at −20°C. The protein pellet was resuspended in 1% RapiGest (Waters, 186002123) and incubated in thermoshaker at 60°C for 15 min at 1050 rpm. The disulfide bonds were reduced by 10 mM dithiothreitol (Thermo Fisher Scientific, Rockford, IL, USA, 20290) (60°C for 45 min at 1050 rpm) and alkylated by 25 mM chloroacetamide (Sigma, Steinheim, Germany, 22790) (37°C for 30 min at 750 rpm). Proteins were then digested by trypsin (Promega, Madison, WI, USA, V5113, 1:20, trypsin to protein ratio) in 50 mM ammonium bicarbonate, pH 8, for 16 hr. Digestion was stopped by addition of 1% formic acid (37°C for 1 hr with shaking at 750 rpm) and the peptide solution was cleared by centrifugation (for 30 min at $21800 \times g$ at 4°C). Obtained peptides were desalted using the C18 extraction disk (Sigma, 66883 U) and dried in vacuum concentrator for MS analysis.

Fusion mass spectrometer (Thermo Fisher Scientific) coupled to Ultimate 3000 HPLC system (Agilent Technologies, Santa Clara, CA, USA) was used for proteomic analysis. Peptides were loaded onto a trap column packed in-house (100 μm ID ×30 mm self-packed with Reprosil-Pur 120 C18-AQ 1.9 μm, Dr. Maisch GmbH, Ammerbuch-Entringen, Germany) and separated at a flow rate of 300 nl/min on an analytical column (75 μm ID ×300 mm self-packed with Reprosil-Pur 120 C18-AQ, 1.9 μm, Dr. Maisch HPLC GmbH). Peptides were eluted from the column with 5–76% linear gradient of increasing buffer B (80% acetonitrile, 0.08% formic acid in water) and decreasing buffer A (0.1% FA in water) with an overall run-time of 90 min. Separated peptides were ionized by electrospray ionization source in a positive ion mode. Full-scan MS spectra were acquired in the range of 350–1550 m/z at a resolution of 60,000 units. The top speed method was selected for fragmentation in the collision cell with Higher-energy Collisional Dissociation with the normalized collision energy of 30% and isolation window of 1.2 m/z.

## Data processing and bioinformatics analysis

Acquired MS spectra were processed using the MaxQuant software package version 1.5.0.25 (*Cox and Mann, 2008*). Spectra were searched using the Andromeda search engine (*Cox et al., 2011*) against the proteome database of *Mus musculus* (Uniprot complete proteome updated at 2014-05-13, with 24,504 entries). MaxQuant search was configured as follows: the mass tolerance was set to 20 and 4.5 ppm for the first and the main peptide search, respectively; the multiplicity was set to one; Trypsin/P was fixed as protease and maximum of 2 missed cleavages were allowed; carbamidomethylation of cysteine was set as fixed modification and methionine oxidation as well as N-terminal acetylation were specified as variable modifications; a false discovery rate of 1% was applied; the re-quantification and match between runs options (Match time window 0.7 min, Alignment time window 20 min) were enabled.

The Protein Groups output file from the MaxQuant was processed by 'Perseus', version 1.5.5.3 for downstream data analysis (*Cox and Mann, 2008*). For each time point, proteins identified in at least two out of four biological replicates in both control and disease group were selected for further analysis. Reverse hits were removed. For Principal Component Analysis (PCA), the LFQ intensities (Label-Free Quantification) were $\log_2$ transformed and averaged by group. PCA was performed in 'Perseus' with number of clusters set to five and Benjamini-Hochberg FDR cut-off of 0.05. For the downstream proteomics analysis, the LFQ intensities of proteins reported by MaxQuant were $\log_{10}$

transformed. The AD/Control intensity ratio for each protein was calculated and $\log_2$ transformed. Proteins with AD/Control ratios showing a statistically significant ($p$-value<0.05; two-sample t-test) fold-change of more than 1.5 or less than 0.667 were selected for further analyses (*Figure 2D–G*). Time course changes in biological pathways and their top upstream regulators were identified by Ingenuity Pathway Analysis (IPA, QIAGEN Inc, https://www.qiagenbioinformatics.com/products/ingenuity-pathway-analysis) (*Figure 2A–C*, *Figure 2—figure supplement 1*). The $\log_2$ AD/Control intensity ratios of all quantified proteins were used for the analysis with IPA. Positive z-score indicates an overall upregulation (activation) of the process, while a negative score stands for its inhibition. The z-score was computed based on the measured protein expression values ($\log_2$ ratio AD/control) and the information on the relationship between the proteins and biological processes they are involved in stored in Ingenuity Knowledge Database.

## Analysis of human mRNA expression datasets

Information on *HEBP1* mRNA expression levels in AD patients was extracted from the transcriptome dataset from the Harvard Brain Tissue Resource Center (HBTRC) that is publicly available on the GeneNetwork website (www.genenetwork.org). Used datasets were human primary visual cortex (GN Accession: GN327), human prefrontal cortex (GN Accession: GN328) and cerebellum (GN Accession: GN326). These datasets were generated on a custom-made Agilent 44K microarray of 39,280 DNA probes uniquely targeting 37,585 known and predicted genes. The study includes 803 participants of which 388 Alzheimer's disease cases, 220 Huntington's disease cases and 195 controls matched for gender, age and postmortem interval.

## Postmortem human brain samples

All experimental protocols were approved and the study conformed to the Code of Ethics of the World Medical Association. All study participants or their legal next of kin gave informed consent and the study was approved by the local ethics committee in Göttingen (No. 24/8/12). All samples were anonymized with regard to their personal data. The brain samples were collected and provided by the Prion Disease Surveillance Units of Germany including spAD, rpAD and non-demented control cohorts as described previously (*Grau-Rivera et al., 2015*; *Zafar et al., 2017a*). Briefly, patient clinical records were retrospectively assessed and classified by two neurologists. Neuropathological assessments were performed by immunohistochemical staining of tissue sections obtained from patients using a selection of antibodies including those directed against β-amyloid and phosphorylated tau. Information on ages, genders, disease duration, disease stage (Braak classification; *Braak and Braak, 1991*) and postmortem interval are summarized in *Table 2*. Brain tissue samples were processed as demonstrated previously (*Grau-Rivera et al., 2015*; *Zafar et al., 2017b*).

## Primary neurons and cell culture

Primary cortical or hippocampal neurons were prepared from postnatal day 0 Wistar rats. Dissected cortices and hippocampi were digested for 30 min with papain enzymatic solution (Worthington Biochemical Corporation, Lakewood, NJ, USA LS003126) in the presence of 1 mg/mL DNaseI (Sigma, D5025). Digestion was stopped by addition of 0.25% BSA (AppliChem, Darmstadt, Germany, A1391) in serum medium (Eagle's MEM (Sigma M2414), 5% FBS (Pan Biotech, Aidenbach, Germany), 2 mM L-alanyl-L-glutamine (Millipore, Berlin, Germany, K0302), 1 × MEM Vitamine (Sigma K0373), Mito+Serum extender (Corning Costar, Kennebunk, ME, USA, 355006) supplemented with 3.8 g/L D-glucose). Digested tissues were triturated using a fire-polished Pasteur pipette until no visible tissue debris could be observed. The cell suspension was passed through a 40 µM cell strainer (Corning Costar, 352340) and subsequently centrifuged for 5 min at 500 rpm followed by resuspension of cell pellets in serum medium. Next, cortical neurons were plated in plating medium (DMEM/F12 (Sigma, D6421), 1 × B-27 (Invitrogen, Rockford, IL, USA, 17504044), 2 mM L-Alanyl-L-Glutamine) directly on poly-D-lysine (PDL) coated 10 cm culture plates (Greiner, Frickenhausen, Germany, 664160) for Co-IP analysis (one cortex per plate). Hippocampal neurons were plated on PDL-coated coverslips in plating medium at a density of 20,000 cells/cm$^2$ for imaging. Medium was changed completely to fresh plating medium supplemented with 1 × FUDR (Sigma, F0503) the next day.

## Immunoblotting

Protein samples were mixed with 4 × NuPAGE LDS Sample Buffer (Thermo Fisher Scientific, NP0008) and boiled at 95℃ for 5 min. 15 µg of protein sample were typically loaded on the gel. Mouse brain samples and Co-IP samples were separated on 12% SDS-PAGE gels. Proteins were transferred to nitrocellulose membrane using the Mini Trans-Blot Cell system (Bio-Rad, Hercules, CA, USA) for 1 hr at constant voltage of 100 V. Human samples were run on 4–20% Criterion TGX gels (Bio-Rad, 5671095) and transferred using Trans-Blot Turbo Transfer System (Bio-Rad). Membranes were blocked in 5% non-fat dry milk in TBST (150 mM Tris-HCl, pH 7.4, 1.5 M NaCl, 0.5% Tween20) for 1 hr at room temperature, incubated with primary antibodies in blocking solution overnight, washed with TBST (five times, 5 min each), incubated with secondary antibodies for 1 hr at room temperature and washed again with TBST (five times, 5 min each). Protein bands were visualized using fluorescence or enhanced chemiluminescence with images developed using Odyssey CLx Infrared Imaging System (Licor, Bad-Humburg, Germany) or Fujifilm LAS-100 device, respectively.

The following primary antibodies were used for immunoblotting: rabbit Hebp1 (1:1000, Invitrogen, PA5-30609), mouse Glo1 (1:1000, GeneTex, Irvine, CA, GTX628890), rabbit CA1 (1:250, Novus Biologicals, Abingdon, UK, NBP1-88191), mouse α-tubulin (1:5000, Synaptic Systems, Göttingen, Germany, 302 211), rabbit β-actin (1:5000, Synaptic Systems, 251 003), rabbit GFP (1:5000, Synaptic Systems, 132 002), mouse Rab5 (1:1000, Synaptic Systems, 108 111), rabbit Rab6 (1:1000, Synaptic Systems, 273 003), rabbit Lamp1 (1:500, Abcam, ab24170), mouse Mic60 (1:1000, Abcam, ab110329), rabbit Cox4 (1:1000, Synaptic Systems, 298 002), rabbit $Cyt_C$ (1:1000, Cell Signaling, Beverly, MA, USA, 11940S), rabbit caspase 9 (1:1000, Abcam, Cambridge, UK, ab185719), mouse Sodium Potassium ATPase, subunit α1 (1:1000, Abcam, Cambridge, UK, ab7671), mouse syntaxin 1 (1:1000, Synaptic Systems, 110 001), mouse VAMP2 (1:10000, Synaptic Systems, 104 211), rabbit phospho-tau (Ser400;Thr403;Ser404) (1:1000, Cell Signaling, Beverly, MA, USA, 11837S). Secondary antibodies against rabbit or mouse were conjugated either with IRDye (Licor) or HRP (Bio-Rad).

## Immunohistochemistry

Mouse tissue samples for immunohistochemistry were prepared as described previously (*Rabe et al., 2012*). In brief, for the preparation of cryosections, one half of the dissected whole brain was fixed in 4% PFA for 4 hr at 4℃ and washed in PBS three times for 20 min each. Tissues were immersed in 15% sucrose in PBS (1 hr), followed by 30% sucrose in PBS (overnight) and finally in 50% tissue freezing medium (Tissue Tek, Leica) in 30% sucrose for 1 hr. Tissue was embedded in the freezing medium, frozen at −20℃ and preserved at −80℃ until use.

For paraffin sections, whole brains of 12-month-old mice were fixed in 4% PFA overnight, washed in PBS three times (20 min each) and subsequently immersed in 0.98% NaCl for 1 hr. The tissues were then dehydrated in a stepwise series of ethanol dilutions (50%, 70%, 90%, 95%, 100%), cleared in the ascending toluene/isopropanol dilution series and finally embedded in paraffin.

Immunostainings were performed on 10 µm thick cryo-sections. Sections were washed three times in PBS and blocked in 10% FCS and 0.5% Triton-X100 in PBS for 60 min at room temperature. Slides were incubated overnight with primary antibodies at 4℃ in blocking solution followed by three washes in PBS (10 min each) and incubation with secondary antibodies (1: 750) for 60 min at room temperature. Finally, sections were rinsed in PBS three times (10 min) and mounted with Vectashield containing DAPI (Vector Laboratories, Burlingame, CA, USA, VEC-H-1500). Additional 8 µm thick paraffin sections were used for IBA1/Hebp1 co-staining. Prior to the staining, paraffin sections were hydrated through descending ethanol series and boiled for one minute in unmasking solution (1:100 in water, Vector Laboratories).

The following primary antibodies were used for immunostainings: rabbit Hebp1 (1:100, Invitrogen, PA5-30609), rat Ctip2 (1:100, Abcam, ab18465), mouse GFAP (1:500, Synaptic Systems, 173 011), mouse GFAP (1:250, Sigma, C9205), guinea pig IBA-1 (1:100, Synaptic Systems, 134 004) and guinea pig NeuN (1:200, Synaptic Systems, 266 004). Secondary antibodies against rabbit, mouse, rat or guinea pig were conjugated either with Alexa488 or Alexa594 dye and were acquired from Invitrogen, Carlsbad, CA, USA. Images were acquired by Leica TCS SP5 confocal laser scanning or Zeiss Axio Vert.Z1 epifluorescent microscope.

Aβ plaques were stained with thioflavin S (Santa Cruz, Dallas, TX, CAS 1326-12-1), as described previously (*Martinez Hernandez et al., 2018*).

## Lentiviral transduction

For overexpression of Hebp1 in primary rat neurons, cDNA encoding full-length human Hebp1 (Origene, RC201873) was subcloned into the FUGW backbone (FUGW was a gift from David Baltimore, Addgene plasmid #14883) using EcoRI and AgeI restriction enzymes. Empty FUGW vector was used as a negative control for overexpression of EGFP. For production of lentiviral particles, HEK293 cells were co-transfected with the FUGW-Hebp1/FUGW plasmid, and the helper plasmids psPax2 (a gift from Didier Trono, Addgene plasmid #12260) and pCMV-VSV-G (a gift from Bob Weinberg, Addgene, plasmid #8454) in a 2:1:1 ratio using Lipofectamine 2000. Medium was changed 6 hr after transfection to DMEM supplemented with 2% FBS and 5 mM sodium butyrate. Culture supernatant was harvested 24 hr and lentiviruses concentrated by ultracentrifugation via Amicon Ultra-15 filters (Millipore, UFC910024). Concentrated lentiviruses were diluted to the final volume of 1 mL in DMEM/F12 medium, aliquoted, snap-frozen in liquid nitrogen and stored at −80˚C until use. Only lentivirus preparations resulting in transduction rate of at least 90% (assessed by EGFP overexpression) were used for experiments.

## Live imaging of mitochondria

Primary rat hippocampal neurons were infected with lentiviruses overexpressing Hebp1-EGFP one day after seeding and analyzed at DIV14. Briefly, cells were incubated with MitoTracker Red CMXRos (Life Technologies, Rockford, IL, USA, M5712) in plating medium at final concentration of 10 nM for minimum of 30 min. Cells were then imaged in Tyrode's solution (10 mM Hepes, pH 7.3, 130 mM NaCl, 4 mM KCl, 5 mM CaCl$_2$, 1 mM MgCl$_2$, 48 mM glucose) using Zeiss Observer 1 laser scanning confocal microscope within 30 min period.

## Identification of Hebp1 binding partners

Proteins interacting with Hebp1 were identified using co-immunoprecipitation coupled with mass spectrometry in four independent biological experiments. Primary rat cortical neurons were infected with lentiviruses expressing Hebp1-EGFP or EGFP one day after seeding. Neurons were lysed at DIV14 with NP-40 lysis buffer. Lysates were clarified by centrifugation at 13,000 × $g$ for 10 min at 4˚C. Hebp1-EGFP and EGFP were pulled down using GFP-trap according to manufacturer's instructions. Beads were sequentially washed in lysis buffer containing descending concentrations of NP-40 (1%, 0.8%, 0.4%, 0.2%). Proteins were eluted by boiling the beads at 95˚C for 10 min in 1 × NuPAGE LDS Sample Buffer and separated on 4–12% Bis-Tris NuPAGE gels (Thermo Fisher Scientific, NP0342). Gels were stained with Coomassie solution overnight and destained in deionized water for two days. Each lane was cut into six equal pieces and in-gel protein digestion was performed as described previously (*Shevchenko et al., 1996*). Peptides extracted from each gel piece were measured three times in independent technical repetitions.

The digested peptides were subjected to Q Exactive HF mass spectrometer (Thermo Fisher Scientific) coupled with an Ultimate 3000 RSLC system (Dionex, USA). Peptides were separated on a self-made capillary column (ReproSil-Pur 120 C18-AQ, 1.9 µm, Dr. Maisch GmbH, 300 × 0.075 mm; C18 pre-column from Thermo Fisher (160454)) with a 5–42% linear gradient of increasing buffer B (80% ACN, 0.08% FA) and decreasing buffer A (0.1% FA in water) for an overall run time of 58 min at a constant flow rate of 300 nl/min. Separated peptides were ionized by electrospray ionization source in a positive ion mode. Full-scan MS spectra were acquired in the range of 350–1600 m/z at the resolution of 60,000 units. The top 30 most abundant precursors were selected for fragmentation in the collision cell with Higher-energy Collisional Dissociation with the normalized collision energy of 30% and isolation window of 1.6 m/z. Max quant search was performed with the same parameters used for brain proteome analysis.

Perseus software was used for downstream data analysis. The intensities of identified proteins were log$_2$ transformed and the missing values for identified proteins in each replicate were imputed with the width of 0.3 and downshift of 1.8 in the total matrix mode. Log$_2$ difference between Hebp1-EGFP and EGFP samples was calculated for each identified protein and was averaged between technical and biological replicates. Statistical significance of protein enrichment in each sample was determined by one-sample t-test ($p < 0.05$).

## Generation of Hebp1 knockout neurons by CRISPR/Cas9 system

sgRNAs (5'-CCCAGCATGGTGACGCCGTG-3' (KO1); 5'-TGGCAGGTTCTAAGCACCGG-3' (KO2); 5'-CCGGTGCTTAGAACCTGCCCA-3' (KO3)) targeting rat Hebp1 were designed using sgRNA Designer (Broad Institute). The pLenti-CRISPR-Hebp1-KO vectors were generated by inserting the sgRNAs into the LentiCRISPRv2 plasmid at the BsmBI site. The LentiCRISPRv2 plasmid is a gift from Feng Zhang (Addgene, plasmid #52961). To obtain Hebp1 knockout neurons, cells were infected individually with lentiviruses generated from the respective pLenti-CRISPR-Hebp1-KO constructs targeting distinct rat sequences of Hebp1 (KO1, KO2 and KO3 sgRNAs) and subsequently tested for Hebp1 expression by immunoblotting. Control neurons were infected with lentiviruses generated from pLenti-CRISPR-Luc vector containing sgRNA targeting luciferase. Control, knockout and wild-type neurons were utilized for further experiments.

## Cell viability and apoptosis assays

Neurons were seeded on black 96-well plates (Corning Costar, 3603) at a density of $7 \times 10^3$ cells/$cm^2$. Cell toxicity was assessed using MultiTox-Glo reagent (Promega, G9270) according to the manufacturer's instructions. Briefly, cytotoxicity was measured by activity of dead-cell protease (luminescent readout) and was normalized to cell viability measured by the activity of live-cell protease (fluorescent readout) to account for discrepancies in cell number between the wells. Final cytotoxicity values are presented as a fold change of corresponding vehicle control. Cells were treated with 10 μM hemin (Sigma, 51289), 25 μM *tert*-butyl-hydroperoxide (Sigma, 458139) or 1 μM staurosporine (Santa Cruz, sc-3510). Hemin was always freshly prepared in accordance with a protocol published previously (Atamna et al., 2015).

Apoptosis was assessed by measurement of caspase 3/7 activity using CellEvent Caspase-3/7 Green Detection Reagent (Sigma, C10723). Briefly, Caspase-3/7 Green Detection Reagent was added simultaneously to cells exposed to hemin alone, 10 μM Aβ$_{42}$ oligomers alone or hemin together with Aβ$_{42}$ oligomers. Cells were fixed and quantified 24 hr later. Aβ$_{42}$ oligomers (Abcam, ab120301) were freshly prepared according to a published protocol (Ryan et al., 2013). Images were acquired from five non-overlapping fields of each well with 20 × objective, Zeiss Axio Observer Z1 Microscope equipped with a motorized stage (DAPI, GFP and Cy3 channel). Cells positive for Caspase-3/7 activity (GFP-positive) were quantified manually as a proportion of total number of cells (DAPI-positive). Each experiment was performed at least in three independent biological repetitions with three technical replicates for each condition.

Mitochondrial membrane potential was assayed with the Mitochondrial Membrane Potential Assay kit (Cell Signaling, 13296) according to the manufacturer's guidelines. Briefly, potentiometric fluorescent TMRE dye (final concentration 200 nM) was added to neurons twenty-four hours after hemin treatment. Fluorescence was measured 30 min after with Tecan reader (Infinite 200 PRO series) plate reader using 550 nm and 615 nm excitation and emission filters, respectively.

## Mitochondria isolation

Isolation of respective crude mitochondrial fractions from wild-type, Hebp1-deficient and control rat hippocampal neurons was followed using a published protocol (Wieckowski et al., 2009). Briefly, neurons were seeded at a density of $1 \times 10^5$ cells in T-25 flasks (Corning, C7046). After lentiviral exposure for 7 days, neurons were washed with PBS prior to trypsin treatment for 2 min. Following cell detachments, respective neurons were spun down and resulting cell pellets were resuspended in PBS for another centrifugation. Corresponding cell pellets were then dissolved in a buffer (225 mM mannitol, 75 mM sucrose, 0.1 mM EGTA and 30 mM Tris-HCl; pH, 7.4) and homogenized using a Teflon pestle (Sigma, Singapore, P7734) with 20 strokes. Respective homogenates obtained were spun down for supernatant collection. Further centrifugation of the supernatants at a higher speed gave rise to respective cytosolic (supernatant) and mitochondrial portions (pellet), which were lysed and subjected to protein analyses.

## Statistical analysis

Proteomic data were analyzed as described in the section 'Data processing and bioinformatics analysis'. Statistical analysis of in vitro cell culture assays and immunoblotting data was performed in Prism

(Graph Pad). Applied statistical tests for each experiment are mentioned in the corresponding figure legends.

## Acknowledgements

We would like to express our sincere gratitude to Prof. Reinhard Jahn for his generous support and invaluable discussions. We thank Brigitte Barg-Kues, Qu Yinghua, Ina Ott, Sigrid Schmidt, Monika Raabe, Annika Kühn and Joana Puchta for their excellent technical assistance. We also thank Dr. Ulrike Teichmann, Thomas Gundlach, Sascha Krause and the entire team of animal caretakers from MPI BPC animal facility for their constant help with animal handling. The research leading to these results was supported by funding from the Deutsche Forschungsgemeinschaft (grant no. CH 1385/1–1) and National University of Singapore. OY and MK-N were supported by scholarships from International Max Planck Research School for Molecular Biology.

## Additional information

### Funding

| Funder | Grant reference number | Author |
|---|---|---|
| Deutsche Forschungsgemeinschaft | CH 1385/1-1 | John JE Chua |
| National University of Singapore | | John JE Chua |
| Max-Planck-Gesellschaft | Open-access funding | John JE Chua |
| International Max Planck Research School for Molecular Biology | Scholarship | Oleksandr Yagensky Mahdokht Kohansal-Nodehi |

The funders had no role in study design, data collection and interpretation, or the decision to submit the work for publication.

### Author contributions

Oleksandr Yagensky, Data curation, Formal analysis, Validation, Methodology, Writing—original draft, Writing—review and editing; Mahdokht Kohansal-Nodehi, Saravanan Gunaseelan, Data curation, Formal analysis, Validation, Writing—review and editing; Tamara Rabe, Data curation, Validation, Writing—review and editing; Saima Zafar, Inga Zerr, Henning Urlaub, Resources, Writing—review and editing; Wolfgang Härtig, Resources, Data curation, Writing—review and editing; John JE Chua, Conceptualization, Resources, Formal analysis, Supervision, Funding acquisition, Validation, Investigation, Methodology, Writing—original draft, Project administration, Writing—review and editing

### Author ORCIDs

Mahdokht Kohansal-Nodehi https://orcid.org/0000-0002-3898-5197
John JE Chua https://orcid.org/0000-0002-5615-1014

### Ethics

Human subjects: All experimental protocols were approved and the study conformed to the Code of Ethics of the World Medical Association. All study participants or their legal next of kin gave informed consent and the study was approved by the local ethics committee in Göttingen (No. 24/8/12). All samples were anonymized with regard to their personal data.

Animal experimentation: All animal procedures used here fully comply with the guidelines as stipulated in the section 4 of the Animal Welfare Law of the Federal Republic of Germany (section 4 of TierSchG, Tierschutzgesetz der Bundesrepublik Deutschland) or in accordance with the Principles of Laboratory Animal Care, and approved by the Institutional Animal Care and Use Committee of the National University of Singapore (protocol number: 2015-01121 (R15-1121)). Procedures performed in the animal facility at the Max-Planck-Institute for Biophysical Chemistry, Göttingen, Germany were

registered accordingly to the section 11 Abs. 1 TierSchG as documented by 39 20 00_2a Si/rö, dated 11th Dec 2013 ("Erlaubnis, Wirbeltiere zur Versuchszwecken zu züchten und zu halten"), by the Veterinär- und Verbraucherschutzamt für den Landkreis und die Stadt Göttingen and examined regularly by the supervisory veterinary authority of the Landkreis Göttingen. All procedures were supervised by the animal welfare officer and the animal welfare committee of the Max-Planck-Institute for Biophysical Chemistry, Göttingen, Germany established accordingly to the TierSchG and the regulation about animal used in experiments, dated on 1st Aug 2013 (TierSchVersV, Tierschutz-Versuchstier-Verordung).

### Decision letter and Author response
Decision letter https://doi.org/10.7554/eLife.47498.030
Author response https://doi.org/10.7554/eLife.47498.031

## Additional files

### Supplementary files
• Transparent reporting form
DOI: https://doi.org/10.7554/eLife.47498.022

### Data availability
All data generated or analysed during this manuscript are included in the manuscript and supporting files. Source data files have been provided for Figs 2, 6, 7, 8 and 9.

The following previously published datasets were used:

| Author(s) | Year | Dataset title | Dataset URL | Database and Identifier |
|---|---|---|---|---|
| Benes F, Schadt E | 2011 | HBTRC-MLC Human Visual Cortex Agilent (Jun11) mlratio | http://www.genenetwork.org/webqtl/main.py?FormID=sharinginfo&GN_AccessionId=327 | GeneNetwork Accession, GN327 |
| Benes F, Schadt E | 2011 | HBTRC-MLC Human Prefrontal Cortex Agilent (Jun11) mlratio | http://www.genenetwork.org/webqtl/main.py?FormID=sharinginfo&GN_AccessionId=328 | GeneNetwork Accession, GN328 |
| Benes F, Schadt E | 2011 | HBTRC-MLC Human Cerebellum Agilent (Jun11) mlratio | http://www.genenetwork.org/webqtl/main.py?FormID=sharinginfo&GN_AccessionId=326 | GeneNetwork Accession, GN326 |

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
