## [Decision Letter]

Thank you for submitting your article "Increased expression of heme-binding protein 1 early in Alzheimer's disease is linked to neurotoxicity" for consideration by *eLife*. Your article has been reviewed by three peer reviewers, one of whom is a member of our Board of Reviewing Editors, and the evaluation has been overseen by Huda Zoghbi as the Senior Editor. The following individuals involved in review of your submission have agreed to reveal their identity: Michael R Kreutz (Reviewer #2).

The reviewers have discussed the reviews with one another and the Reviewing Editor has drafted this decision to help you prepare a revised submission.

Summary:

This is a nice study that is motivated to gain a better understanding of the early stages of Alzheimer's disease (AD). Yagensky et al. have searched for potential presymptomatic markers by examining early brain proteome changes in an AD mouse model relative to age-matched controls. The authors identify heme-binding protein-1 (Hebp1) as a novel dysregulated protein, which is highly elevated in 3×Tg mice by 6 months of age at a time when synaptic plasticity defects are known to become apparent; Hebp1 is also found elevated in brains of patients with rapidly progressive forms of AD. Notably, Hebp1 knock-down greatly attenuates heme-induced cytotoxic cell death, suggesting that elevated levels of Hebp1 exacerbate heme-dependent cell death. Overall, the main strengths of the study are the identification of Hebp1 as a novel protein implicated in neurodegeneration in AD and determining a neuronal function for Hebp1 in mediating heme-induced cytotoxicity. While these findings are of general interest, there are some issues that warrant careful consideration. A compelling experimental support for the claim of Hebp1 as a presymptomatic marker for AD is lacking, and the causal relationship between elevated Hebp1 and neurodegeneration in AD remains to be established. Moreover, there are technical weaknesses in data presentation/description/statistical analysis in the manuscript as it stands.

Essential revisions:

In order to relate Hebp1 dysregulation to neurodegeneration phenotype of 3×Tg AD mouse model/AD patients the following points need to be addressed:

1) Test whether overexpression of Hebp1 in neurons increases heme-induced cytotoxicity.

2) Test whether knock-down of Hebp1 expression is neuroprotective against Aβ-induced cytotoxicity.

3) Determine if heme is elevated in 3×Tg AD mice. The proposed role for Hebp1 as a presymptomatic marker for AD is not necessarily consistent with its role in cell death that occurs at later stages of AD. A stronger evidence for dysregulation of Hebp1 at earlier stages needs to be shown (or otherwise do not refer to Hebp1 as a presymptomatic marker for AD):

4) Perform western blots of Hebp1 in control and 3×Tg-AD mice brains at 2 months of age using at least 3 animals per group, with the expectation of elevated Hebp1 in 3×Tg mice relative to controls.

5) Perform western blots for Hebp1 in brains from control and those suffering from mild cognitive impairment.

Additional technical issues that require careful consideration are as follows:

6) Figure 5. Panels A,B – There is no substantial enrichment in Hebp1 and relatively big contamination with tubulin. One cannot conclude a specific association with mitochondria based on this data. Panels C-H – High-resolution imaging and/or negative control should be included to demonstrate that the co-localization pattern of Hebp1-GFP and Mitotracker is not random. In panel E, where is the area shown at higher resolution taken from? There seems to be appreciable number of Hebp1-GFP puncta that are not associated with mitochondria. Additional markers for endosomes and the plasma membrane in the subcellular fractionation and/or images would be helpful.

7) The authors refer to a previous publication with patient information. However, it would be helpful to have this information in this manuscript. Can the authors provide data on APOE4 status? How were the individuals diagnosed with AD? Was AD confirmed pathologically? What was the postmortem interval for the brains?

8) Please provide the N's for each experiment. For example:

- In Figure 1—figure supplement 1, it is clear 4 individual brains were analyzed for pTau. It is also clear that 9 control, 8 slow progressing and 8 rapid progressing AD brain were analyzed for Hebp1 levels. But, beyond that, it is hard to tell what the N's are. Does replicate mean individual brains or 1 brain split into 4 samples?

- Figure 5 is missing the information about the number of replicates.

9) Appropriate statistical tests should be performed.

- For the proteomics data, why was a Student's test performed and not a 2 factor ANOVA with genotype and time as factors?

- In Figure 3, a Student's t test after an ANOVA is not appropriate; a Bonferroni correction is required to account for multiple comparisons.

- In Figure 3—figure supplement 2, why does the data for Hebp1 transcripts appear so bimodal – in both the controls and AD cases. How were the statistics performed? A Student's t test would not be appropriate here since the data does not appear to be normally distributed.

10) The relevant brain region used for the experiment needs to be described:

- Figure 1—figure supplement 1C – from which brain region the extracts have been prepared?

- Figure 4B – which cortical area?

11) Figure 4. For images of the hippocampus, zoomed out images should be shown and also at higher resolution. In panel C, it would be helpful to have higher magnification images to show Hebp 1 localization in neurons and not in glia.

[Editors' note: further revisions were requested prior to acceptance, as described below.]

Thank you for resubmitting your work entitled "Increased expression of heme-binding protein 1 early in Alzheimer's disease is linked to neurotoxicity" for further consideration at *eLife*. Your revised article has been favorably evaluated by Huda Zoghbi (Senior Editor), a Reviewing Editor, and one reviewer.

The manuscript has been greatly improved with additional experiments and careful editing. However, the following issue needs to be addressed before acceptance.

Figure 5. The claim for specific association of Hebp1 to mitochondria is still not compellingly supported by the imaging data shown. To compare the relative distributions of Hebp1-EGFP vs. soluble EGFP with respect to Mitotracker, panels E and F should present data for at least 7 microns, similarly to panel D, and all in the same scale. The text subsection “Hebp1 interacts with mitochondrial contact 240 site complex” should be toned down to reflect the data. The images show that Hebp1 "can be" associated to mitochondria, and whether a "substantial proportion" of Hebp1-EGFP is closely juxtaposed to Mitotracker remains to be justified. Is there a reason for not assessing the subcellular localization of endogenous Hebp1 using immunofluorescence labelling (as has been done for hippocampal sections)?

---

## [Author Response]

Essential revisions:In order to relate Hebp1 dysregulation to neurodegeneration phenotype of 3×Tg AD mouse model/AD patients the following points need to be addressed:1) Test whether overexpression of Hebp1 in neurons increases heme-induced cytotoxicity.

As suggested, we evaluated whether Hebp1 overexpression could increase heme-induced cytotoxicity in neurons. Briefly, DIV1 hippocampal neurons were infected with lentiviruses expressing either EGFP alone or Hebp1-EGFP. Seven days after transduction, the neurons were treated with 5 or 10 μM of hemin or vehicle only. Cell toxicity was assessed 24 hours after the treatment using the RealTime-Glo assay (Promega). The results shown below were obtained from 6 independent experiments.

The results indicate that more cytotoxicity was present in Hebp1-EGFP expressing neurons as compared to EGFP expressing neurons for each hemin concentrated used. However, significantly more cytotoxicity was observed when hemin concentrations were increased from 5 to 10 μM within each group. These results support the CRISPR-Cas9 data showing that Hebp1 expression is required for heme-mediated cytotoxicity while indicating that endogenous levels of Hebp1 are sufficient to mediate heme-induced cytotoxicity. The extent of cytotoxicity depends, instead, on the concentration of heme that the neurons are exposed to. This finding is highly significant given that vascular damage progressively worsens over the course of AD progression which, in turn, contribute to rising heme levels that will ultimately contribute to progressive neuronal loss (Ghiso et al., 2010). Nevertheless, we still do not completely understand why elevation of Hebp1 levels occurs at the early phase of the disease. We have included these results as supplemental data (Figure 7—figure supplement 1) and have modified the text accordingly to describe and discuss these results (subsection “Hebp1 facilitates heme-mediated cytotoxicity”).

2) Test whether knock-down of Hebp1 expression is neuroprotective against Aβ-induced cytotoxicity.

We exposed control and Hebp1-KO hippocampal neurons to Aβ42 with or without co-exposure to hemin and evaluated cytotoxicity by quantifying the population of caspase 3^+^/7^+^ cells. Exposure of wildtype and LUC neurons to Aβ42 triggers cytotoxicity in approximately 40% of the neurons. Co-exposure of neurons to heme and Aβ42 further elevated this to approximately 88%, indicative of an additive effect. Remarkably, Hebp1-KO neurons consistently demonstrated significantly lower cytotoxicity in all 3 treatments (heme;~9%, Aβ42;~7%, Aβ42+Heme;~23%). This demonstrates that knockdown of Hebp1 also protects against Aβ-induced cytotoxicity. Since Aβ42 also induces neuronal apoptosis, these results suggest that Hebp1 participates in shared pathways triggered by Aβ and heme that cause neuronal cell death. These new results have been inserted as Figure 9 in the revised manuscript. Additional text has also been inserted to describe and discuss these results (subsection “Hebp1 facilitates heme-mediated cytotoxicity”).

3) Determine if heme is elevated in 3×Tg AD mice.

We agree with the reviewers that determination of increased heme would be an important issue to address. However, detection of heme elevation during the early (asymptomatic) phase of AD in the brains of these animals, which would be pertinent to this study, is difficult. Such minute early aberrations are likely to be confined mostly within the local microenvironment and below the detection limit of most assays. Moreover, low levels of heme at this stage would probably be masked by contaminating heme leftover from microscopic quantities of residual red blood cells (RBCs) or from partial lysis of RBCs during tissue preparation. Although more refined methods for detecting and measuring heme have been reported, these techniques are not readily available to us and are unfortunately not possible to establish within the given time frame.

Nevertheless, we have highlighted that cerebral amyloid angiopathy contributes significantly to pathology in the 3×Tg-AD model starting from the early stage of the disorder (Li and Praticò, 2015). Furthermore, pathophysiological changes in brain vascular have also been reported in these animals by other independent studies, correlating with those seen in afflicted humans (Grammas et al., 2014; Lin et al., 2014). Hence, we reason that heme would also be elevated in these mice. We have modified the Discussion section to include the observations on vascular pathology in 3×Tg-AD mice.

We hope these modifications and considerations now better clarify the relationship between Hebp1 dysregulation and neurodegeneration.

The proposed role for Hebp1 as a presymptomatic marker for AD is not necessarily consistent with its role in cell death that occurs at later stages of AD. A stronger evidence for dysregulation of Hebp1 at earlier stages needs to be shown (or otherwise do not refer to Hebp1 as a presymptomatic marker for AD):4) Perform western blots of Hebp1 in control and 3×Tg-AD mice brains at 2 months of age using at least 3 animals per group, with the expectation of elevated Hebp1 in 3×Tg mice relative to controls.

We thank the reviewers for highlighting this and would like to note that these data were indeed included in previous manuscript as Figure 2K. Four biological replicates each from 2-month-old control and 3×Tg-AD mice were presented. We apologise for not indicating this clearly and have revised the figure legend to highlight this more prominently.

5) Perform western blots for Hebp1 in brains from control and those suffering from mild cognitive impairment.

We have discussed this issue extensively with several collaborators including Dr Oliver Wirths (University Medical Center, Goettingen), an AD expert. While we agree with the reviewers that it will be insightful to look into Hebp1 levels in MCI patients, it is extremely difficult to obtain postmortem samples from these patients for technical reasons. The samples have to be obtained from patients already included in a neurodegenerative disorder study (for clinical documentation of MCI) that have expired due to non-cerebral pathological causes and this has to have happened close to the inclusion date. The chances for this to coincide are very slim and such patients were not available to us in this study.

Collectively, we hope that these modifications and responses (together with those listed earlier) have helped provide stronger evidence for dysregulation of Hebp1 at the earlier stages, and to better reconcile the observation of Hebp1 as an early AD marker with its role in causing progressive neuronal death in AD. Indeed, our hypothesis is that an early increase in Hebp1 could serve a neuroprotective function by, for example, eliminating small populations of damaged neurons. However, subsequent more massive activation of this mechanism (e.g. via progressive accumulation of excessive heme over a larger area at later stages of AD) would then contribute to the progressive neuronal loss during these later phases. Hence, early elevation of Hebp1 (and thus by definition a presymptomatic marker) would not be necessarily inconsistent with its role in contributing to cell death at later stages.

Additional technical issues that require careful consideration are as follows:6) Figure 5. Panels A,B – There is no substantial enrichment in Hebp1 and relatively big contamination with tubulin. One cannot conclude a specific association with mitochondria based on this data. Panels C-H – High-resolution imaging and/or negative control should be included to demonstrate that the co-localization pattern of Hebp1-GFP and Mitotracker is not random. In panel E, where is the area shown at higher resolution taken from? There seems to be appreciable number of Hebp1-GFP puncta that are not associated with mitochondria. Additional markers for endosomes and the plasma membrane in the subcellular fractionation and/or images would be helpful.

We thank the reviewers for highlighting this and agree that there is indeed a noticeable amount of tubulin in Figure 5A and B. Nevertheless, we wish to draw the reviewers’ attention to the data presented in Figure 8A and B where mitochondrial isolation experiments were also performed, albeit using an improved procedure. Here, the substantial enrichment of Hebp1 in the mitochondrial (Mt) fraction versus the cytosol (S) fraction obtained from control untreated neurons can be readily observed with only minute quantities of cytosolic Hebp1 (e.g. Figure 8A left panel). Importantly, no tubulin contamination of the mitochondrial fraction was detected. Likewise, mitochondrial markers CytC and Cox4 were only detected in the mitochondrial fraction. In the revised manuscript we have updated Figure 5B (previously Figure 5A) with results obtained from the modified mitochondria isolation procedure to show that Hebp1 is indeed substantially enriched in mitochondrial fractions and that tubulin is not detectable in these preparations. We have also additionally probed for markers of the endosomal, lysosomal, synaptic and plasma membrane compartments (Rab5, Rab6, LAMP1, VAMP2 and Syntaxin 1) in our mitochondrial preparations using the improved procedure. We confirm that Hebp1 is present in the mitochondrial fraction and that the aforementioned markers from other cellular compartments could not be detected in this fraction. The immunoblot for these data is presented as Figure 5B in the revised manuscript.

We note that the subcellular fraction protocol employed in Figure 5A (previously Figure 5B) was optimized for the isolation of synaptic components which likely accounted for the higher contamination of tubulin in the crude mitochondrial fraction. Accordingly, we have revised the text to clarify this issue.

For the immunofluorescence experiments, we have now included an EGFP only negative control (Figure 5C top panel) and the corresponding line scan (Figure 5D) to show that association of Hebp1-EGFP with mitochondria is not random as the distribution of EGFP alone shows no correlation with Mitotracker. While we agree that Hebp1 can also be present in other organelles, we hope that these data have convincingly supported our conclusion that a notable fraction of Hebp1 is associated with mitochondria.

7) The authors refer to a previous publication with patient information. However, it would be helpful to have this information in this manuscript. Can the authors provide data on APOE4 status?

We thank the reviewers for highlighting this. With regards to the postmortem intervals for the brains, this information was listed in the right-most column in the table (“Postmortem delays”) formerly provided as a supplement to Figure 3, now as Table 2.

We apologize for the insufficient data detailing patient information in the previous manuscript. To address the reviewers’ concerns, the relevant information has been extracted from the publications (Grau-Rivera et al., 2015; Zafar et al., 2017).

How were the individuals diagnosed with AD?

Two neurologists retrospectively reviewed the clinical records and registered the date of disease onset (defined as the first neurological or psychiatric symptom reported in the clinical records that could be attributed to the underlying pathological process) and final clinical diagnoses, based on the treating physician’s best clinical judgment.

Was AD confirmed pathologically? What was the postmortem interval for the brains?

Neuropathological diagnoses were established according to current standard diagnostic criteria by two neuropathologists. Immunohistochemistry was performed in selected brain areas using specific antibodies that include those directed against β-amyloid (Dako, clone 6F/3D, dilution 1: 400), phosphorylated tau (Thermo Scientific, clone AT8, dilution 1: 200), ubiquitin (Dako, polyclonal, dilution 1: 400), α-synuclein (Novocastra, clone KM51, dilution 1: 500), TDP43 (Abnova, clone 2E2-D3, dilution 1: 500) and prion protein (Millipore, clone 3F4, dilution 1: 300). The level of neuropathological changes caused by AD was classified as low, intermediate or high, according to the current National Institute of Aging-Alzheimer’s Association (NIA-AA) guidelines. These guidelines include the assessment of the topographic distribution of β-amyloid deposits (A for amyloid, a scale of 1–3, 1: cortical and limbic, 2: 1 + basal ganglia, and 3: 1 + 2 + midbrain and cerebellum), a simplification of Braak’s staging system of neurofibrillary pathology (B for Braak, a scale of 1–3, 1: transentorhinal, 2: limbic, and 3: neocortical) and a semiquantitative assessment of neuritic plaques as suggested by the Consortium for Establishing a Registry for Alzheimer Disease (C for CERAD, a scale of 1–3, 1: sparse, 2: moderate, and 3: frequent).

In the revised version, we have included a summary of this information in the subsection “Post mortem human brain samples”, reproduced below:

“Briefly, patient clinical records were retrospectively assessed and classified by two neurologists. Neuropathological assessments were performed by immunohistochemical staining of tissue sections obtained from patients using a selection of antibodies including those directed against βA4-amyloid and phosphorylated tau.”

Unfortunately, the ApoE4 status of the patients are not available to us.

8) Please provide the N's for each experiment. For example:- In Figure 1—figure supplement 1, it is clear 4 individual brains were analyzed for pTau. It is also clear that 9 control, 8 slow progressing and 8 rapid progressing AD brain were analyzed for Hebp1 levels. But, beyond that, it is hard to tell what the N's are. Does replicate mean individual brains or 1 brain split into 4 samples?- Figure 5 is missing the information about the number of replicates.

We apologize for not clearly specifying the N’s. This information is now included in the revised manuscript. Replicates refer to individual brains (i.e. biological replicates).

9) Appropriate statistical tests should be performed.- For the proteomics data, why was a Student's test performed and not a 2 factor ANOVA with genotype and time as factors?

For proteomics data, we have used Student's t-test as the aim of this study was to compare relative protein expression levels between control and 3×Tg-AD mice at each specific time point. We agree that two-factor ANOVA would be more appropriate, if conclusions on statistically significant differences in protein expression between mice of different ages and genotypes were to be assessed.

- In Figure 3, a Student's t test after an ANOVA is not appropriate; a Bonferroni correction is required to account for multiple comparisons.

We thank you and the reviewers for highlighting this. We have reanalyzed data with one-way ANOVA followed by Bonferroni’s multiple comparisons test for individual pairs of samples and modified the figure accordingly. The differences remain statistically significant.

- In Figure 3—figure supplement 2, why does the data for Hebp1 transcripts appear so bimodal – in both the controls and AD cases. How were the statistics performed? A Student's t test would not be appropriate here since the data does not appear to be normally distributed.

The source data for this figure was derived from 3 datasets prepared by Harvard Brain Tissue Resource Center (HBTRC) that are publicly available on the GeneNetwork website (www.genenetwork.org). To better represent the data spread, we have now replotted the data using violin plots in the revised manuscript that show the datasets are not bimodal.

We have also confirmed that a normal distribution was indeed not observed for some of the data series. Following the reviewers’ recommendations, we have now reanalyzed the data with Mann-Whitney test and modified the figure and accompanying legend accordingly. The differences observed remain statistically significant.

10) The relevant brain region used for the experiment needs to be described:- Figure 1—figure supplement 1C – from which brain region the extracts have been prepared?- Figure 4B – which cortical area?

We apologise for omitting this information. In Figure 1—figure supplement 1C, the soluble fractions were obtained from one of the sagittal half of the harvested mouse whole brains. In Figure 4B, the top panels were imaged from the fronto-temporal cortex depicting primary motor and somatosensory areas.

11) Figure 4. For images of the hippocampus, zoomed out images should be shown and also at higher resolution. In panel C, it would be helpful to have higher magnification images to show Hebp 1 localization in neurons and not in glia.

We have now included zoomed out images of the hippocampus from 24-month-old control and 27-month-old 3×Tg-AD mice. We have also included higher magnification images supporting the expression of Hebp1 in neurons but not glial. These images have been included as supplemental data to Figure 4 (Figure 4—figure supplements 1 and 2, respectively).

[Editors' note: further revisions were requested prior to acceptance, as described below.]

The manuscript has been greatly improved with additional experiments and careful editing. However, the following issue needs to be addressed before acceptance.Figure 5. The claim for specific association of Hebp1 to mitochondria is still not compellingly supported by the imaging data shown. To compare the relative distributions of Hebp1-EGFP vs. soluble EGFP with respect to Mitotracker, panels E and F should present data for at least 7 microns, similarly to panel D, and all in the same scale.

We have now included the results of the analyses to include data for 7 microns presented in the same scale for Figure 5D, E and F as suggested by the reviewers.

The text in subsection “Hebp1 interacts with mitochondrial contact 240 site complex” should be toned down to reflect the data. The images show that Hebp1 "can be" associated to mitochondria, and whether a "substantial proportion" of Hebp1-EGFP is closely juxtaposed to Mitotracker remains to be justified.

As suggested, we have toned down the text of the respective section. It now reads as follows:

“We further demonstrated that the protein can be mitochondrially-associated by detecting the presence of Hebp1 from mitochondria isolated from cultured hippocampal neurons (Figure 5B). Supporting the biochemical data, we observed that a portion of EGFP-tagged Hebp1 expressed in rat primary neurons appears to be closely juxtaposed to mitochondria (visualized using Mitotracker) (Figure 5C and line scans in Figures 5E and 5F).”

Is there a reason for not assessing the subcellular localization of endogenous Hebp1 using immunofluorescence labelling (as has been done for hippocampal sections)?

The anti-Hebp1 antibody did not work well for immunocytochemistry (no strong specific signals were detectable), unlike for immunohistochemical sections. We have tried various blocking conditions and antibody concentrations without success.